# Probiotics in the Sourdough Bread Fermentation: Current Status

Ingrid Teixeira Akamine, Felipe R. P. Mansoldo and Alane Beatriz Vermelho *

Bioinovar, Institute of Microbiology Paulo de Góes, General Microbiology, Federal University of Rio de Janeiro—UFRJ, Rio de Janeiro 21941-902, Brazil
* Correspondence: abvermelho@micro.ufrj.br

**Abstract:** Sourdough fermentation is an ancient technique to ferment cereal flour that improves bread quality, bringing nutritional and health benefits. The fermented dough has a complex microbiome composed mainly of lactic acid bacteria and yeasts. During fermentation, the production of metabolites and chemical reactions occur, giving the product unique characteristics and a high sensory quality. Mastery of fermentation allows adjustment of gluten levels, delaying starch digestibility, and increasing the bio-accessibility of vitamins and minerals. This review focuses on the main steps of sourdough fermentation, the microorganisms involved, and advances in bread production with functional properties. The impact of probiotics on human health, the metabolites produced, and the main microbial enzymes used in the bakery industry are also discussed.

**Keywords:** sourdough bread; microorganism; enzymes; probiotics; fermentation; microbiome

## 1. Introduction

Sourdough fermentation is a technique that can use several types of flour, such as wheat, rye, or other cereals, and water. The oldest process of sourdough preparation is spontaneous fermentation and acidification due to the local microbiota in a complex interaction, mainly lactic acid bacteria (LAB) and yeasts [1–3]. It is a traditional fermentation process used for various foods, especially baked goods, improving nutritional and health benefits [4,5]. The best-known example is bread; however, other applications of this type of fermentation include biscuits, panettone, pasta, and beverages [6–9]. Several varieties of this traditional sourdough bread recipe have appeared, bringing innovations to the primary process to attend to the bakery industry and the consumer, which appreciates nutritious, healthy, and tasty food [10]. Sourdough can be used by the bakery industry in several applications through liquid, pasty, or dry formats with live microorganisms. Moreover, there are "ready-to-use" sourdoughs with microorganisms inactivated to promote shelf-stable products [11].

When sourdough is used, rather than *Saccharomyces cerevisiae*, a typical baker's yeast, for the bread dough fermentation, superior properties in bread quality and technological value are achieved [12].

Probiotics are safe for human consumption and can produce metabolites that positively influence gastrointestinal [13] and bone health diseases [14]. Beneficial effects in eczema [15], allergies [16], respiratory tract infection [17], obesity [18], and cognitive and mental health are also described [19]. Probiotics act as gut microbiota, stimulating a complex communication with the body that is mediated by the metabolites generated [20]. Some LAB and yeasts present in sourdough fermentation are considered and presumed to be probiotics [21]. However, a complete analysis, such as a whole-genome analysis, is proposed for better probiotic characterization [22–26].

This study aims to review improvements in sourdough fermentation, its enzymes, metabolites, and technological advances to produce sourdough bread with desirable properties.

**2. Sourdough Fermentation Types: Inoculum and Technology Processes**

Sourdough fermentation can be classified according to the inoculum (type I, II, and III) and the technology process (type 0, I, II, and III). According to inoculum, type I uses the microorganisms present in the dough, which results in spontaneous fermentation and utilizes back-slopping to propagate the sourdough microbiota. This process is common in artisan bakeries and homes. Control parameters include temperature, number, and time of back-slopping in artisan bakeries. Type II inoculums are frequently utilized in industrial processes, where a starter culture is added to sourdough fermentation according to the objectives and results desirable in the final process. Type III is a hybrid and combines type II by using a starter culture with type I for propagating the sourdough with back-slopping. Type III is common in artisan bakeries and industrial ones [27].

According to the production and process, sourdough fermentation can also be classified into four major types. The simple type 0 fermentation starts with a flour–water mixture and is allowed for a limited time. LAB species that are naturally present in the flour, grow faster are more prolific than yeast. Aplevicz et al. 2013 [28] compared the growth of two *Lactobacillus paracasei* strains with two *Saccharomyces cerevisiae* strains. LAB had the highest counts over time and microbial growth after 10 h of fermentation. The highest value for LAB strains was 8.91 log CFU/g, and then 8.66 log CFU/g. One *S. cerevisiae* strain reached 8.03 log CFU/g, and the other, 0.21 log CFU/g. At 6 h of fermentation, LABs were 8.66 log CFU/g and 8.33 log CFU/g. Yeasts showed 7.18 log CFU/g and 7.08 log CFU/g.

The growth kinetics of six strains of LAB: *Fructilactobacillus sanfranciscensis*, were compared with five strains of yeast: *Kazachstania humilis*, from sourdough, by Altilia et al. 2021 [29]. Three predictive models were used to evaluate the behavior of co-cultivated microorganisms: The zwitwering model based on Gompertz's equation, Baranyil and Roberts' function, and Schiraldi's function. The results showed that *F. sanfranciscensis* strains significantly steer the growth kinetics affecting the ratio of bacterial to yeast cells, where the yeast strains of *K. humilis* adapt to the bacterial strains. The authors discuss the possibility of metabolic interactions for stabilizing the sourdough consortium through communication, such as quorum sensing, to control population density, among other functionalities. In another co-culture article with LAB and yeast, the yeast population size decreased in the presence of LAB regardless of the strain, while the LAB's population size was rarely influenced by the presence of yeast [30]. In type 0 fermentation, bioactive molecules and organic acids (lactic and acetic acids) are produced lowering the pH (pH~4), but it is controversial if this type of fermentation is a true sourdough. The time limit is insufficient to produce other characteristic products of sourdoughs and is more known as sponge dough [27].

Type I is a traditional method for preparing sourdough and consists of spontaneous fermentation in a flour–water mixture at an ambient temperature of less than 30 °C for 24 h or less and back-slopping frequently. This fermentation increases the quality of the final baked good [27,31].

Type II is when a starter culture initiates the fermentation process and is used to develop desirable characteristics for the baking industry. It is a sourdough in liquid form with controlled parameters [31,32].

Type III can be dried or lyophilized. It is preferable for use in an industrial bakery because it has a higher stability quality than fresh sourdough [31,33]. The fermentation temperature used is above 30 °C and in a single time stage between 24–72 h [34].

The fermentation process will change depending on the added materials, such as the addition of naturally rich products in microorganisms: fruits, yogurt, or another material [35]. It is observed that it is challenging to classify sourdough fermentation due to some nuances in the process production [34]. Figure 1 summarizes the major types of production and process of and their characteristics.

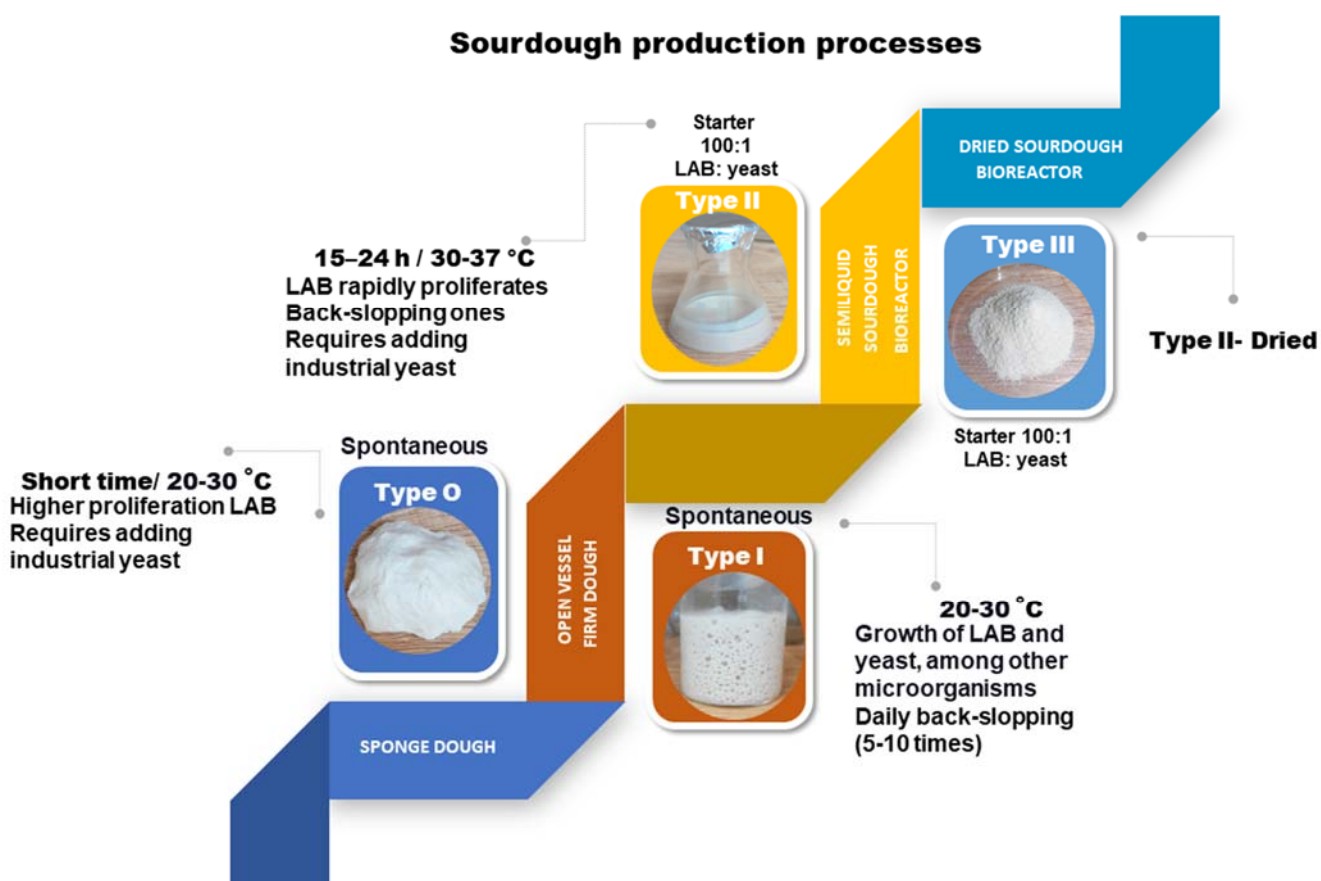

**Figure 1.** Sourdough production of the four types of processes. Type II and III can be scaled to an industrial level.

### 3. Sourdough Fermentation: Major Pathways

The sourdough fermentation process generates mainly acids, alcohols, aldehydes, esters, and ketones; it is the primary route of volatile organic compounds (VOCs) [36]. Sourdough bread has a complex profile, strongly influenced by the compounds generated during fermentation by the diverse array of microorganisms present, mainly yeasts and LAB.

Wheat flour is composed of starch (70–75%), water (14%), proteins (10–12%), non-starch polysaccharides (2–3%), lipids (2%), and soluble carbohydrates such as maltose, sucrose, and glucose (1.55–1.85%) The starch is degraded into glucose and maltose by flour amylase and glucoamylase by some sourdough LABs. Maltose is degraded into glucose by the enzyme maltose phosphorylase from LAB and by alpha-D-glucosidase, a maltase from *Saccharomyces* yeasts. In the dough, the available carbon source is thus maltose, followed by sucrose, glucose, and fructose, with some trisaccharides such as maltotriose and raffinose [37].

The glucose concentration increases during fermentation because other complex carbohydrates are metabolized by LAB and yeasts. The disaccharide lactose can be fermented by microbial enzymes liberating glucose and galactose after cleavage. Starting from glucose, homofermentative LAB produces lactic acid through glycolysis, while heterofermentative LAB generates, besides lactic acid, $CO_2$, acetic acid, and/or ethanol [38]. Two major fermentation routes are found: lactic fermentation (LF) and alcoholic fermentation (AF). In lactic fermentation, the pyruvate molecules formed by glucose oxidation from the Embden–Meyerhof–Parnas or glycolytic pathway are reduced to lactic acid (homolactic fermentation). *Streptococcus*, *Lactobacillus*, and *Enterococcus* use this route (Figure 2A1). Another possibility is the pyruvate originating from a mixture of lactate, ethanol and/or acetic acid, through

the oxidation of the coenzymes NADH + and H+ by the lactate dehydrogenase, and $CO_2$ from a glucose molecule until ribulose 5 phosphate (heterolactic fermentation).

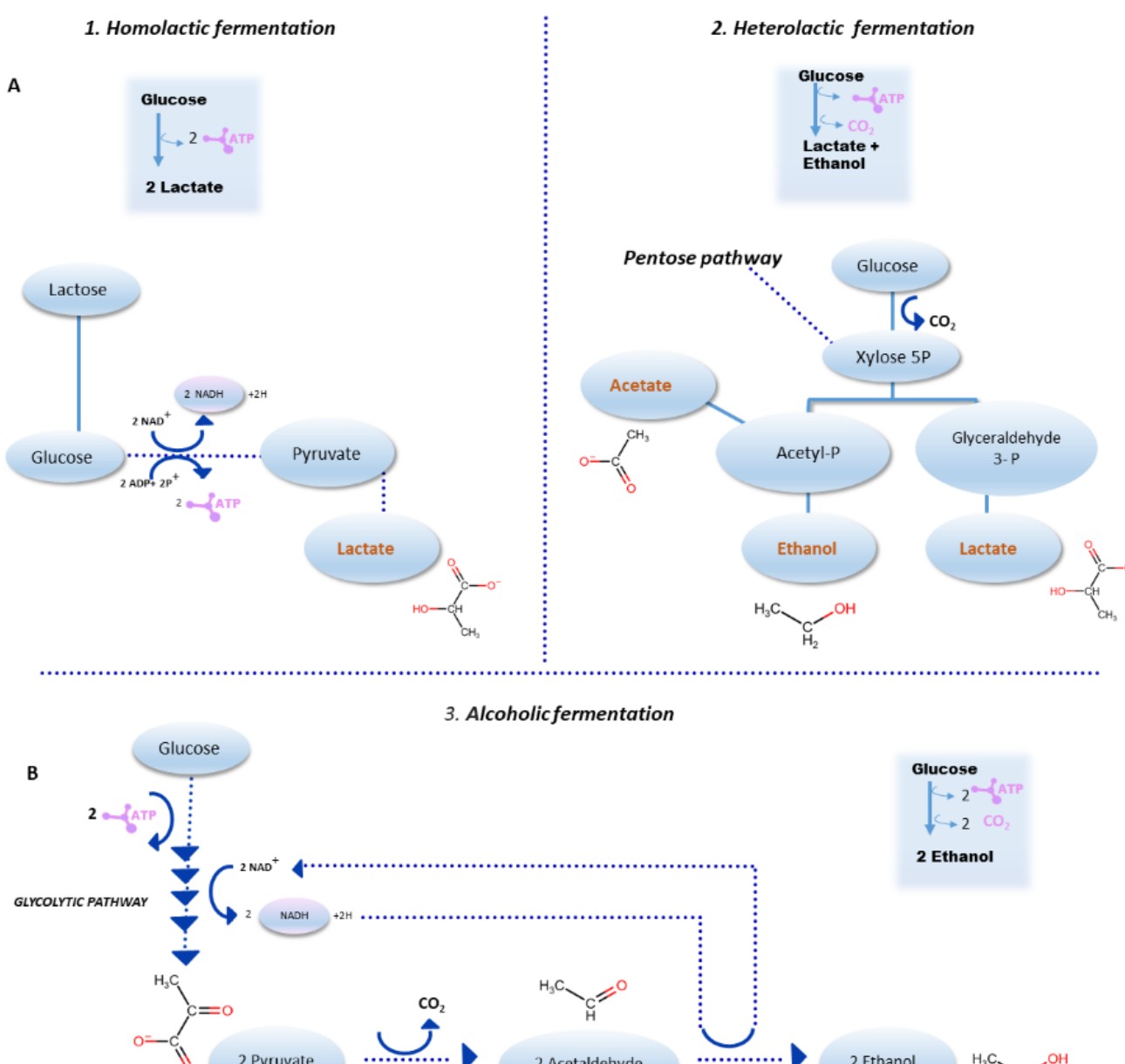

**Figure 2.** Major sourdough fermentation pathways. (**A**) Lactic fermentation. (**B**) Alcoholic fermentation. (**A1**), homolactic fermentation; (**A2**), heterolactic fermentation. (**B3**), alcoholic fermentation from yeast. The formation of ethanol and $CO_2$ from the reduction of pyruvate characterizes alcoholic fermentation.

This fermentation is used by genera such as *Leuconostoc*, *Bifidobacterium*, *Weissella*, and some *Lactobacillus*. Some glycolytic enzymes are missing in these bacteria, and they use the pentose-P pathway to degrade the glucose (Figure 2A2). Classification according to the type of lactic fermentation defines three major *Lactobacillus* groups: group I: obligate homofermentative lactobacilli. Hexoses are exclusively fermented to lactic acid via the Embden–Meyerhof–Parnas pathway. The bacteria have the enzyme fructose-1,6-bisphosphate aldolase, but phosphoketolase is absent. Due to this, pentoses and gluconate are not fermented. This group includes the species *L. acidophilus*, *L. delbrueckii*, and *L. salivarius*; group II: facultative heterofermentative lactobacilli. Hexoses are fermented to lactic acid almost exclusively via the Embden–Meyerhof–Parnas pathway. The bacteria

possess both aldolase and phosphoketolase enzymes, and ferment not only hexoses but also pentoses. In the presence of glucose, the enzymes of the phosphogluconate pathway are inhibited. This group includes *L. casei, L. paracasei, L. plantarum,* and *L. curvatus.* Organic acids, $CO_2$, alcohol, and $H_2O_2$ could be produced; group III: composed by the obligate heterofermentative Lactobacilli. They have the enzyme phosphoketolase but not aldolase and ferment sugars in a heterofermentative mode. Hexoses are fermented via the phosphogluconate pathway, producing lactate, ethanol (acetic acid), and $CO_2$ in equimolar amounts. Pentoses enter this pathway and can also be fermented [39–42].

After the enzymatic hydrolysis of pentosans and other complex carbohydrates, pentoses such as d-xylose, ribose, and l-arabinose are liberated in rye and wheat flour. Heterofermentative strains can ferment pentoses through part of the 6-phosphogluconate pathway [43]. Pentoses and hexoses are simultaneously rather than successively fermented by sourdough Lactobacilli. Lactobacilli are important during the fermentation of sourdoughs, and they can ferment pentose carbohydrates without producing $CO_2$ because they have a constitutive phosphoketolase. Facultatively heterofermentative lactic acid bacteria produce the phosphoketolase enzyme in response to the presence of pentoses [42,44]. The contribution of LAB to the flavor of sourdough bread is associated with the production of lactic acid (fresh acidity) and acetic acid (sharp acidity). The conversion of amino acids such as phenylalanine (sweet), isoleucine (acidic), glycine, serine, and alanine (vinegar/sour) to aldehydes and ketones can form additional flavor compounds [36].

Alcoholic fermentation (AF) is characterized by the formation of ethanol and $CO_2$ from pyruvate reduction (Figure 2B). AF is a metabolic pathway of yeasts. In this fermentation, yeast produces gas that promotes dough conditioning and increased volume and imparts desirable aromas and flavors in baked goods. A synergy occurs between the dough and the yeast, and in this context, the rate of $CO_2$ production is determined by the activities of the glycolytic yeast enzymes. The retention of $CO_2$ produced by yeast is a function of the wheat or cereal flour. It is essential to point out that glucose formed by LAB amylases from amide is a significant source of this substrate for yeasts. Other compounds are volatile organic compounds (VOCs), which arise from this interaction between yeasts and LABs. Diverse aldehydes, alcohols, esters, ketones, lactones, sulfur compounds, furan derivatives, and hydrocarbons are found, conferring flavor and other characteristics to sourdough bread [32]. Some VOCs are due to yeast fermentation, particularly of *S. cerevisiae* in mixed sourdough starters, including ethanol, 2,3-butanedione, 2-methyl-1-propanol, 3-methyl-1-butanol, and phenyl ethyl alcohol [45]. Other negatively correlated compounds have been associated with the fermentation of homofermentative and facultative heterofermentative LAB, including 2,3-butanedione and acetaldehyde [46].

## 4. Probiotics and Postbiotics in Sourdough and the Impact on Human Health

Probiotics are live microorganisms that confer a health benefit when administered in adequate amounts. In addition, new terms have been used to name microbial metabolites from non-viable cells, including paraprobiotics, parapsychobiotics, ghost probiotics, matabiotics, tyndallized probiotics, bacterial lysates, and postbiotics. However, the last term is being mightily used for the vital concept of promoting health. Postbiotics means non-living microorganisms and is a preparation with inanimate microbial cells or their components that promote host health. Metabolites and cellular structures from microorganisms from sourdough are potential postbiotics [47,48].

Sourdough microbiota are involved in a complex interaction that engenders beneficial compounds for human consumption. With the current molecular methods, a better characterization of sourdough microbiota has been described [49,50]. Moreover, whole-genome sequencing has permitted more accuracy in confirming the requirements for characterizing probiotic microorganisms [25].

The probiotics and postbiotics of sourdough provide health benefits in addition to the essential nutritional components of the bread itself. A fermented food passes through a process with microorganisms and enzymes converting macronutrients into bioactive

products. LAB is one major microorganism group with several probiotic strains applicable to cereal fermentation. LAB are classified as Gram-positive bacteria which include low guanine + cytosine (G + C) content as well as being acid-tolerant, non-motile, non-spore-forming, and are rod- or cocci-shaped. The main function of LAB is to produce lactic acid, that is, the acidification [40]. The most common genera found in sourdough were *Lactobacillus*, *Leuconostoc*, *Weissella*, and *Pediococcus*. However, the genera *Streptococcus*, *Enterococcus*, and *Lactococcus* are found, although less frequently [51,52]. The second group of microorganisms are the yeast, and more than 20 species dominated by the genera *Saccharomyces* and *Candida* are found. LABs and yeasts are naturally found in the microbiota of the cereal flour used or can be added as a starter. Some native microorganisms found in sourdough are listed in Table 1.

No strong relation between the microbiota and regional origin has been proven until now [34,53]. However, in Brazil, *Lactobacillus farciminis* was the prevalent species rather than *Lactobacillus sanfranciscensis*. Other bacteria could be present. At a temperature of 30 ± 1 °C, unusual bacterial groups such as *Pseudomonas* and *Enterobacteriaceae* can proliferate. At a temperature of 21 ± 1 °C, there may be a prevalence of LAB, which are safer for human consumption [54].

Other factors that have been identified that could influence the microbiota. Insects could be a reservoir of *Lactobacillus sanfranciscensis* [55], commonly found in type I sourdough. *Lactobacillus delbrueckii* and *Lactobacillus reuteri* groups from vertebrate–host interactions are typical occurrences in type II sourdough [33]. In Italy, bakery LAB species such as *Weissela* spp. and *Streptococcus* spp. are in the local environment [50]. In addition, the water used in sourdough type I could influence its microbiota [56]. It has been demonstrated that *Lactobacillus*, *Streptococcus*, *Enterococcus*, and Lactococcus are endophytic components of wheat and, as such, a source of sourdough microbiota [52]. The sourdough microbiota changes with decreasing pH, whereas at the beginning, there is abundant Proteobacteria phylum in mature sourdoughs, which at the end is replaced by Firmicutes [54,57]. De Angelis et al. [58] demonstrated that the extraction rate of wheat flour from cultivars *Triticum aestivum* and *Triticum durum* modified the microbiota community in sourdough type I. The study showed that during fermentations, both controlled and uncontrolled pH conditions up to 4.0 yield similar LAB and yeast species [59].

A meta-analysis with 583 sourdough-related literature articles in the period 1999–2017 was done by Kerrebroeck et al. 2017 [34] based on the study of species diversity (LAB and yeasts) and process condition (type I and II). The authors concluded that the species number of yeasts is lower than LABs and the microbial diversity of type I sourdoughs is lower than type II. Through the principal component (PC) analysis, three clusters were detected. The first group had a high relative abundance of *Pichia kudriavzevii* and/or *Torulaspora delbrueckii*. In the second group, a high relative abundance of *Wickerhamomyces anomalus* and/or *Candida glabrata* was established, and the third *Saccharomyces cerevisiae* and/or *C. humilis* species were detected.

A systematic analysis from 1990 to 2020 using "sourdough" as a keyword showed the following number of articles by geographic location: North America (13), South America (6), Africa (31), Europe (175), Asia (54), China (20), and Oceania (1—New Zealand). In Latin America and America, very few studies have been published. *Lactobacillus plantarium* and *Saccharomyces cerevisiae* appeared in several sourdough microbiota characterizations [1]. It also showed that some bacteria not belonging to the LAB group could significantly function in sourdough and need to be explored more. Some species are listed in Table 1. On the other hand, in a study that characterized 500 samples from sourdough microbiota, almost all from the United States of America (429 samples) showed evidence of different bacterial groups of the Rhodospirillales order despite the Lactobacillales being the dominant order. *Lactobacillus sanfranciscensis* was the prevalent species. It was demonstrated that the acetic acid bacteria group substantially impacted aroma and leavening properties in sourdough fermentation [53].

A sourdough starter is in essence, a culture of probiotic yeast and lactic bacteria to be added to the fermentation process, besides the native microbiome of the cereals and external contaminations.

Starters also can be used to propagate the sourdough as a live fermentation. In this context, it is interesting to explore the technological and biological properties of the selected starter associations of bacteria/yeast to improve the quality of fermentation and bread properties. Acidification and proteolysis analysis (including free amino acids), and volatile compound profiles are examples. In a recent review of Landis et al. [53], working with 500 sourdough starters from four continents, it was suggested that geographical location has little influence on the microbial diversity of bacteria and yeast species. This result contrasts sharply with widespread assumptions relative to the biogeographic distribution of starters from sourdough [34]. In Brazil, for instance, *Lactobacillus farciminis* was the prevalent species rather than *Lactobacillus sanfranciscensis*. Table 2 lists some starters used in sourdough and produced compounds.

**Table 1.** Native microbiota found in sourdough.

| | Microbiota | Reference |
|---|---|---|
| Bacteria | *Acetobacter lovaniensis* spp.<br>*Acetobacter malorum* ssp.<br>*Acetobacter pasteurianus/papaya*<br>*Acetobacter tropicalis*<br>*Enterecoccus durans*<br>*Enterobacter hormaechei/cloacae*<br>*Enterococcus faecalis*<br>*Enterococcus faecium*<br>*Enterococcus gilvus*<br>*Enterococcus hirae*<br>*Gluconobacter frateurii* spp.<br>*Gluconobacter sphaericus* spp.<br>*Komagataeibacter cluster 4*<br>*Lactobacillus brevis*<br>*Lactobacillus coryniformis*<br>*Lactobacillus curvatus*<br>*Lactobacillus diolivorans*<br>*Lactobacillus farciminis*<br>*Lactobacillus fermentum*<br>*Lactobacillus gallinarum*<br>*Lactobacillus kimchii*<br>*Lactobacillus otakiensis*<br>*Lactobacillus parabrevis*<br>*Lactobacillus paracasei*<br>*Lactobacillus paralimentarius*<br>*Lactobacillus plantarium*<br>*Lactobacillus sakei*<br>*Lactobacillus sanfranciscensis*<br>*Lactobacillus xiangfangensis*<br>*Lactococcus lactis*<br>*Leuconostoc*<br>*Leuconostoc citreum*<br>*Pediococcus*<br>*Pediococcus parvulus*<br>*Pediococcus pentosaceus*<br>*Psychrobacter*<br>*Streptococcus*<br>*Weissella* | [1,34,53,54,60,61] |

**Table 1.** *Cont.*

| Microbiota | Reference |
|---|---|
| Yeasts | |

In the Microbiota column (Yeasts row):

*Cida glabrata*
*Cida humilis*
*Hanseniaspora uvarum*
*Kazachstania humilis (synonym Cida humilis)*
*Kazachstania servazzii*
*Kazachstania unispora*
*Kluyveromyces aestuarii*
*Kluyveromyces lactis*
*Kluyveromyces marxianus*
*Pichia fermentans (synonym Cida lambica)*
*Pichia kudriazevii*
*Saccharomyces cerevisiae*
*Saccharomyces uvarum*
*Saccharomycestales* sp.
*Torulaspora delbrueckii*
*Wickerhamomyces anomalus*
*Yarrowia keelungensis*

Reference: [1,34,53,61–63]

**Table 2.** Starters used in sourdough bread production with identified compounds.

| Inoculum | Compound | Property | Reference |
|---|---|---|---|
| *Lactobacillus sanfranciscensis* and *Lactobacillus reuteri* | Glutaminase activity | Significantly influences wheat bread flavor | [64] |
| *Lactobacillus diolivorans* and *Lactobacillus buchneri* | Propionate | Increase antifungal property on bread | [65] |
| *Weissella cibaria* MG1 | Dextrans | Improved volume, crumb softness, and shelf-life | [66] |
| *Lactobacillus hammessi* | Monohydroxy fatty acid | Antifungal property improving the bread shelf-life | [67] |
| *Lactobacillus paracasei* RN5, *Lactobacillus plantarum* X2, *Lactobacillus brevis* LBRZ7, *Lactobacillus fermentum* LBRH10, *Lactobacillus buchneri* LBRZ6, and *Propionibacterium frendenreichii* ssp. *Shermanii* NBIMCC 327 | Antimicrobial | Prevent bacterial and mold spoilage | [68] |
| *Lactobacillus curvatus* 750(13), *Pediococcus acidilactici* EKO26, *Pediococcus pentosaceus* 1850(3), *Lactobacillus coryniformis* pA, *Weissella cibaria* EKO31, *Pediococcus pentosaceus* EKO23, *Lactobacillus plantarum* KKp 593/p, *Lactobacillus helveticus* 10, *Lactobacillus plantarum* W37/54, *Lactobacillus sakei* 750(20), and *Lactobacillus rhamnosus* Lr (23) | Proteolytic activity | Reducing allergenic proteins and improving the quality of bakery products | [69] |
| *Lactobacillus curvatus* MA2, *Pediococcus pentosaceus* OA2, and *Pediococcus acidilactici* O1A1 | Phytase and antioxidant activities | Improve textural and sensory features of bread | [70] |
| *Weissella cibaria* PON10030, *Weissella cibaria* PON10032, *Lactobacillus citreum* PON 10079, and *Lactobacillus citreum* PON10080 | Volatile organic compounds | Improve the taste of bread | [71] |
| *Saccharomyces bayanus* | Aromatic compounds | Improve the sensory profile of bread | [72] |
| *Propionibacterium freudenreichii* | B12 Vitamin | Potential improvement in nutritional and health value of bread | [73] |
| *Torulaspora delbrueckii* | Aromatic compounds | Improve the sensory profile of bread | [72] |
| *Lactobacillus reuteri* LTH5448 and *Lactobacillus reuteri* 100-23 | γ-glutamyl dipeptides | Influence in the salty taste | [74] |

**Table 2.** *Cont.*

| Inoculum | Compound | Property | Reference |
|---|---|---|---|
| *Lactobacillus sanfranciscensis*, *Candida milleri*, and transglutaminase | Isodipeptide bonds, ketones, medium-chain fat acids, and alcohols | Positive effects on bread rheological features, shelf-life, and aroma profile | [75] |
| *Lactobacillus acidophilus* ATCC20552 and *Bifidobacterium lactis* Bb 12 | Antimicrobial | Inhibit rope-forming *B. subtilus* | [76] |
| *Kluyveromyces marxianus* and *Saccharomyces cerevisiae* | Inulinase | Reduction of fructans, consequently FODMAPs in dough prepared with whole wheat flour | [77] |
| *Kluyveromyces marxianus* | Inulinase, phytase | Reduction of FODMAPs Highest porosity and lowest hardness | [62,78] |
| *Saccharomyces cerevisiae* and *Pediococcus pentosaceus* | Phytase activity | Phytic acid decrease | [79] |
| *Gluconobacter oxydans* IMDO A845 | Higher amount of lactic acid | Positive aroma profile of sourdough bread | [80] |
| *Leuconostoc citreum* FDR241 | glycosyltransferase | dextran concentration in sourdough | [81] |
| *Enterococcus mundtii* QAUSD01 | Proteolytic activity | Gluten-degrading | [60] |
| *Wickerhamomyces anomalus* QAUWA03 | Proteolytic activity | Gluten-degrading | [60] |
| *Lactobacillus plantarum* | Phenolic acid esterase, decarboxylases, reductase, and wide range of glycosil hydrolases | Its influence in bread quality needs study | [33] |
| *Weissella cibaria* VTT E-153485 | Peptidase | Increased proteolysis in faba bean dough | [82] |
| *Weissella confusa* VTT E-143403 | Dextran | Increased viscosity in faba bean dough | [82] |
| *Pediococcus pentosaceus* VTT E-153483 and *Leuconostoc kimchi* VTT E-153484 | Phytase | Reduction of phytic acid in faba bean dough | [82] |
| *Lactobacillus amylovorus* DSM19280 6% and *Weisella cibaria* MG1 18% | Organic acid and exopolysaccharide | Low-salt bread with desirable shelf-life, and high sensory quality (volume and crumb texture) | [83] |
| *Lactobacillus brevis* and *Lactobacillus plantarum* at 35 °C | Volatile compounds | Improve texture and aromatic properties of sourdough bread | [84] |
| *Lactobacillus reuteri* | Organic acid, saturated fatty acid, hydroxy fatty acid | Anti-aflatoxigenic capability and antifungal activity | [85] |
| *Lactobacillus plantarum* 29DAN and *L. plantarum* 98A | Polyphenol | Antioxidant activity | [86] |
| *Lactobacillus plantarum* NOS7315, *Lactobacillus rossiae* NOS7307, *Lactobacillus brevis* NOS7311, and *Saccharomyces cerevisiae* PS7314 | Synergistic fermentation | Improved bread sensory characteristic | [87] |
| *Bacillus licheniformis* | Exopolysaccharides (EPS) | Immunomodulatory potential | [88] |
| *Lactobacillus paracasei* K5 | Organic acid, higher the complexity of volatile compounds | Decrease spoilage, increase shelf-life, and improve sensory properties in sourdough bread | [89] |
| *Enterococcus mundtii* QAUSD01 and *Wickerhamomyces anomalus* QAUWA03 | Proteolytic activity | Toxic gliadin degraded in the sourdough fermentation | [90] |
| *Lactobacillus plantarum* CH1 | Antifungal compounds | Do not interfere in the sensory quality of bread | [91] |
| *Streptococcus thermophilus* | Glucosyltransferase B | Bread with lowly digestible starch and textural improvements | [92] |
| *Pediococcus pentosaceus* SP2 | Organic acid content | Reduce mold and rope spoilage | [93] |
| *Enterococcus faecium* and *Kluyveromyces aestuarii* | High phenolic and antioxidant capacity, respectively | Improve bread quality | [62] |
| *Lactobacillus reuteri* TMW1.656 | Reutericyclin | Inhibition of growth of rope-forming bacilli in bread | [94] |

**Table 2.** *Cont.*

| Inoculum | Compound | Property | Reference |
|---|---|---|---|
| *Wickerhamomyces anomalus* P4 | Phytase | Decrease phytate and increase mineral solubilization in sourdough bread | [95] |
| *Levilactobacillus brevis* TMW 1.211, *Pediococcus claussenii* TMW 2.340 from breweries | O2-substituted (1,3)-β-D-glucan | Improving water binding capacity | [96] |
| *Weissella confusa/cibaria* 3MI3 from sourdough | Dextran | Technological properties of dough and bread, such as water absorption, rheology, stability in cold storage, bread staling, and syneresis of starch gels/avoided the resistant starch formation | [97] |
| *Pediococcus lolii* B72 and *Lactiplantibacillus plantarum* E75 from mature sourdough | Volatiles compounds | Improving sensorial acceptability | [98] |

The microbial production of bioactive peptides, enzymes, organic acids, exopolysaccharides (EPS), and vitamins are considered the primary metabolites responsible for antioxidant, antimicrobial, and probiotic activities [99]. Relative to fermentative processes, many bioproducts are present in sourdough fermentation. Koistinen et al. [100] demonstrated 118 bioactive compounds in sourdough fermentation. Some of these health benefits and others are summarized in Table 3.

**Table 3.** Microorganisms and the compounds they produce, and the health benefits.

| Microorganisms | Compound | Benefit | Reference |
|---|---|---|---|
| Probiotics *Streptococcus thermophilus*, *Lactobacillus plantarum*, *L. acidophilus*, *L. casei*, *L. delbrueckii* spp. *bulgaricus*, *Bifidobacterium breve*, *B. longum*, and *B. infantis* | Peptidase | Alfa-gliadin degradation, reduced wheat allergenic | [101] |
| LAB from sourdough | Gamma-aminobutyric acid (GABA) | ACE-inhibitory activity | [102] |
| LAB from sourdough | Multifactors | Low-glycemic index in the white wheat bread | [103] |
| *Lactobacillus reuteri* | Exopolysaccharide | Antiadhesive properties, inhibition enterotoxigenic *Escherichia coli* | [104] |
| *Lactobacillus brevis* with *S. cerevisiae* var. *Chevalieri*; *L. Fermentum*; *L. Fermentum* with phytase | Higher total phenolic and a lower molar ratio of lactic to acetic acid | Reduce glycemic index | [105] |
| *L. curvatus* SAL33 and *L. Brevis* AM7 | Peptide lunasin | Cancer preventive | [106] |
| *Bifidobacterium* strains | Phytase | Increase iron bioavailability in bread | [107] |
| *Weissella ciabaria* MG1; *L. reuteri* VIP, *L. reuteri* Y2 | Oligosaccharides | Improved nutritional quality of sorghum bread | [108] |
| *L. brevis* | Phytase | Decrease phytate levels, improve mineral bio-accessibility | [109] |
| *L. Sakei* KTU05-6 | Organic acids, bacteriocins | Bio-preservative | [110] |
| *Weissella confusa* LBAE C39-2 | Dextransucrase (glycoside hydrolase) | Alfa-glucans/ oligosaccharides or glycoconjugates | [111] |
| *L. rossiae* DSM 15814 from sourdough | Vitamin B12, folate, and riboflavin | Nutritional improvement | [112] |
| *Lactobacillus amylovorus* DSM 19280 and *Weisella cibaria* MG1 | Glutamate accumulation | NaCl reduction in bread | [83] |
| Traditional sourdough LAB starter culture | Essential and non-essential amino acids, flavonoids, antioxidant peptides | Nutritional improvement protects against oxidative stress and degenerative disease through phenolic compounds | [113] |

**Table 3.** *Cont.*

| Microorganisms | Compound | Benefit | Reference |
|---|---|---|---|
| LAB from traditional Austrian sourdoughs | Fructose metabolized/ antifungal and anti-bacillus properties | Decrease FODMAPS/ control molds | [114] |
| *Lactobacillus plantarum* ZJUFT17 from Chinese sourdough | In mice, decreased: the profile, insulin resistance, lipopolysaccharide, cytokines interleukin (IL)-1β, tumor necrosis factor (TNF)-α | Managing gut microbiota, decreasing pathogenic and pro-inflammatory microbes, and stimulating anti-obesity ones | [115] |
| *Levilactobacillus brevis* TMW 1.211, *Pediococcus claussenii* TMW 2.340 from breweries | O2-substituted (1,3)-β-D-glucan | Prebiotic effect in bread, improving water binding capacity | [96] |
| *Lactobacillus plantarum* ZJUFB2 from Chinese sourdough | Probiotic effect on gut microbiota | Prevent insulin resistance and modulate gut microbiota. | [116] |
| *Levilactobacillus brevis* TMW 1.2112, *Pediococcus claussenii* TMW 2.340 | Dietary fiber, short acid fat chain SCFA, butyrate, propionate,β-glucan | Healthy environment in the colon, chemopreventive | [117] |
| *Pediococcus pentosaceus* F01, *Levilactobacillus brevis* MRS4, *Lactiplantibacillus plantarum* H64, and C48 | Γ-aminobutyric acid (GABA) | Bread from surplus bread with high nutritional value | [118] |
| *Weissella cibaria* PDER21 | α-D-glucan | Antioxidant properties | [119] |

One important point is that most probiotics die when the bread is baked at high temperatures. However, most health benefits continue, mostly, not with probiotics at intestinal epithelial cell colonization but in sourdough dough. It is essential to note that active biomolecules could be present in the baked bread. Different conditions, varying temperatures, baking time, and fermentation parameters could influence the heat resistance of some bioproducts, and they could be present as nutrients and fibers such as β-glucan and resistant starch [120]. Increasingly, the literature discusses this important point. If the baking time is shortened by increasing the baking temperature or reducing the bread size, higher residual viability of LAB species may be obtained after baking [121]. Studies with *L. plantarum* at three different baking temperatures showed a reduction from 8.8 log CFU/g to 4–5 log CFU/G [122]. Concerning yeast, the count decreased from an initial value of 9 log CFU/g to 4 log CFU/g with baking at 200 °C for 13 min [121].

During the baking process, cell lysis of microorganisms delivers cellular debris that may also have beneficial properties functioning as postbiotics. Examples of postbiotic compounds that are present in sourdough are short-chain fatty acids (SCFAs), secreted proteins/peptides, bacteriocins, secreted biosurfactants, amino acids, flavonoids, EPS, vitamins, organic acids, and other molecules discussed in this review. Remains of microbial cell structures such as peptidoglycan from bacterial cell walls, pili, fimbriae, flagella, cell-surface-associated proteins, cell-wall-bound biosurfactants, and cell supernatants are also postbiotic components that have potential health benefits in the host [47,123].

There is both a growing demand from consumers for additive-free, safe, and nutritious foods and for bread with a longer shelf-life and less staling due to microbial spoilage, and sourdough offers some advantages [124].

Several microorganisms, mainly LAB, have antimicrobial properties, which favors their use as probiotics and as a bioprotective culture in fermented products. Koistinen et al. [125] suggest that metabolites with this effect produced by LAB may synergistically modulate the local microbial ecology, such as in the gut. Some strains of LAB can produce a variety of other antimicrobial compounds, such as organic acids (lactic, acetic, formic, propionic, and butyric acids). Organic acids have been described with antifungal actions against *Aspergillus*, *Fusarium*, *Monilia*, and *Penicillium* [126]. Ethanol, fatty acids, enzymes, acetoin, hydrogen peroxide, diacetyl, antifungal compounds (propionate, phenyl-lactate, hydroxyphenyl-lactate, cyclic dipeptides, and 3-hydroxy fatty acids), bacteriocins (nisin,

reuterin, reutericyclin, pediocin, lacticin, enterocin, and others), and bacteriocin-like inhibitory substances (BLIS) are also described in the literature [127].

Bacteriocins are a class of antimicrobial proteins or peptides considered safe for the body and do not induce resistance. This antimicrobial can be applied against harmful food pathogens and used as a food preservative (preserving foods such as dairy products, canned foods, and meats) and a natural antiseptic with bactericidal or bacteriostatic effects. *Lactococcus lactis* is the microorganism producing nisin, the first fully identified bacteriocin [128]. Among lactic acid bacteria, one of the most versatile and adaptable is *Lactiplantibacillus plantarum*, with a long history of use in foods and claims of benefits as a probiotic. This species has shown promising results in protecting post-harvest fruits and controlling mold in foods such as beverages and bread [129]. *Lactiplantibacillus plantarum* strain ITM21B has been used to prolong bread shelf-life due to the production of antimicrobial substances, such as lactic, acetic, phenyl lactic (PLA), and hydroxy-phenyl lactic (OH-PLA) acids [130]. Varsha et al. 2014 [131] demonstrated that the antifungal compounds produced during fermentation resist sterilization temperatures and remain active adding value for the baking industry.

The antimicrobial effect reduces the salt content in bread. Helping reduce sodium consumption to the recommended level of 5–6 g/day will benefit health, particularly regarding blood pressure and cardiovascular disease. Taking into consideration that bread and cereals represent the major sources of salt in the human diet, microbial bioprocessing has been proposed to reduce salt content while keeping the flavor [132].

In addition, the hydrolysis of flour proteins leads to the production of amino acids, organic acids, and other metabolites, improving the nutritional and functional values of sourdough bread [133].

Dietary sourdough bread increases short-chain fatty acid, isovaleric and 2-methyl butyric acids, and γ-aminobutyric acid (GABA) content, among other bioactive molecules [134]. The γ-aminobutyric acid GABA is a non-protein amino acid synthesized from L-glutamate by the glutamate decarboxylase [135]. It is a mammalian neurotransmitter involved in critical regulatory functions, hypotensive properties, anti-depressive effects, diuretic, and antioxidant effects [136]. It is present in many medicines and supplements [137].

Several strains of LAB can carry out activities that promote human health, such as modulating the immune response, preventing cancer, reducing chronic intestinal inflammation, and cholesterol levels. Other functions include improving the intestinal barrier, inhibiting pathogenic organisms, and beneficial interactions with the endogenous intestinal microbiota [138]. These bacteria are present in sourdough and liberate bioproducts in bread dough, such as vitamin B12, folate, and riboflavin [112].

Although probiotics from sourdough are not present in the intestine environment, they provide gastrointestinal benefits, including the pre-digestion of non-nutritional molecules. Sourdough bacteria and yeast decrease the phytic acid present in wheat. The phytate (myo-inositol hexaphosphate) content of whole wheat products is a concern because it inhibits the absorption of minerals, has a negative effect on the nutritional properties, and causes gastrointestinal disorders, digestive discomfort, and flatulence. On the other hand, removing wheat bran and germ reduces the nutritional value of bread. The acidification during sourdough fermentation stimulates endogenous grain phytase, but LAB and yeast phytase activity significantly reduces the phytate content of the bread [124]. Another advantage is the reduction of fermentable oligo-, di-, and monosaccharides and polyols such as sorbitol and mannitol (FODMAPS) and amylase–trypsin inhibitors (ATIs) from wheat. Both are related to trigger intestinal symptoms in irritable bowel syndrome and non-celiac wheat sensitivity. A study by Boakye et al. 2022 [139], described that the longer sourdough fermentation time (12 h) caused up to 69%, 69%, and 41% reductions in fructans, raffinose, and ATIs, respectively. Low FODMAPs may be valuable for people with gastrointestinal disorders.

The reduction of gluten content is important for people with gluten-related disorders. The long fermentation process of sourdough also helps break down gluten proteins

in wheat. Moreover, certain bacteria produce peptidases. Some *Lactobacillus* strains, including *L. sanfranciscensis, L.rossiae, L. plantarum, L. brevis, L. pentosus, L. alimentarius, L. fermentum, L. paracasei, L. casei subsp. casei,* and *P. pentosaceus* degrade gluten. Other gluten-degrading bacteria have been isolated form sourdough, such as *Bacillus* spp. [140], and *Enterococcus mundtii* and the yeast *Wickerhamomyces anomaly* from locally fermented sourdoughs (Khamir) [60]. Gluten has baking properties, but its protein components, gliadin, and glutenins, could trigger undesirable immune responses, causing inflammation and damage [141].

Antioxidant and anti-inflammatory activities of peptides from cooked sourdough breads were described [142], where longer fermentation (72–96 h) increases the production of aromatic compounds with antioxidant activity [143].

LAB and yeast strains were isolated and molecularly identified from traditional Iranian sourdough. Based on total phenol production and antioxidant capacity assessments, all the identified traditional sourdough microorganisms significantly produced phenolic compounds. They showed significant antioxidant capacity improving bread's health benefits and quality [62].

EPS such as β-glucan, dextran, and inulin, are metabolites from LAB identified in sourdough fermentation. The β-glucan, for instance, is a prebiotic homopolysaccharide formed by glucose, presenting substantial health benefits, such as stabilized cholesterol levels, anti-inflammatory effects, and benefits for probiotic microorganisms [96]. The conclusions reached by Schlörmann et al. [117] provide significant insights into the general chemopreventive and prebiotic effects of LAB-generated sourdoughs and bread. In addition, exopolysaccharides are produced by LAB in sourdough, favoring great water retention. β-glucans also contribute to the viscoelastic properties of dough [96]. In the case of thermal treatment at 60 °C and 80 °C for up to 60 min, the molecular mass of the EPS was not significantly affected [144].

The glycemic index (GI) is the speed at which sugar enters our body's bloodstream and this parameter is affected by probiotics from sourdough. It is reported that sourdough bread has a lower GI than yeast bread. Bread with carbohydrates and starches, which are rapidly assimilated, increase the GI. An explanation of the decrease in the glycemic index is the higher level of lactic acid produced under these fermentation conditions. Lactic acid effectively diminishes postprandial glycemic and insulinemic responses, whereas acetic acid brings out a delayed gastric emptying rate. Moreover, the starch availability is reduced under baking heat, thus allowing the breakdown of sucrose to form EPS such as glucans which contribute to the rise in dietary fiber content [145,146]. Mutlu et al. [147] discussed that LAB could contribute to lowering the glycemic index by limiting the digestion rate of starch and converting glucose from digested starch into several bioproducts such as organic acids, fatty acids, sugar alcohols, bioactive peptides, and indigestible exopolysaccharides. Similar results were found by Demirkesen-Bicak et al. [148], where the authors demonstrated results supporting the consumption of wheat and whole wheat bread produced by type 2 fermentation due to higher resistant starch and slowly digestible starch, and lower rapidly digestible starch and glycemic index values showing beneficial effects. However, some controversy exists in the literature. A systematic review selected 18 studies published up to June 2021 in the EMBASE, MEDLINE, Scopus, and Web of Science databases. The authors concluded that the consumption of sourdough bread has a lower influence on blood glucose compared to that of industrial bread or glucose [149].

In addition to the health effects, sourdough brings flavor, aroma, and better texture of bread due to LAB enzymatic hydrolysis processes. Flavor from volatile compounds in sourdough are due to peptidases conversion of glutamine released from cereal proteins to glutamate, and conversion of arginine to ornithine by LAB during baking. Microbial and enzymatic reactions, such as lipid oxidation by cereal enzymes, improve flavor and texture. LAB and yeast in sourdough produce aroma precursors such as free amino acids, which lead to the generation of aldehydes or corresponding alcohols [126,150]. Table 3 summarizes the functionality of several microorganisms studied in this fermentation. It

was demonstrated that sourdough bread fermentation conditions considerably influence bioactive compound viability [100]. Furthermore, many studies have described unique characteristics and molecules formed from the selective use of inoculum as a starter in bread making.

Using a starter culture and varying the temperature and fermentation time changes the flavor and texture in sourdough bread [84]. Moreover, selected LAB and manipulated temperature of sourdough fermentation could increase the β-glucan yield [96]. Methods for maintaining viable starter cells for use in the baking industry have been studied widely, such as immobilization techniques [93]. Studies showed that using the spray drying technique to preserve the viability of lactic acid bacteria and yeasts in the dry mass was much more effective than other drying techniques [5]. However, this area needs further studies since they can vary with the cell type or microbial consortia [151]. Studies have been realized to develop better kinetic control in the spray drying process of LAB, which is fundamental for their growth and viability [152]. Understanding the stress response mechanisms used for LAB can help industries develop better processes to produce LAB at a large scale [153]. Encapsulation of LAB has been demonstrated as a method to improve the viability of LAB during drying processes [154]. For instance, microencapsulation of cells by spray drying with cellulose acetate phthalate was more efficient than with maltodextrin [155]. Using probiotics in bread is a challenge due to the higher temperature of baking. However, a recent study showed that the strain of *Lactobacillus plantarum* P8 survived baking and increased the number of its cells during storage [122].

## 5. Enzymes in Sourdough

Enzymes are responsible for several biochemical events in sourdough fermentation. Although some enzymes are present in the cereal, most of them are produced by microorganisms. From this perspective, the selection of native yeasts in sourdough for use in the baking industry has increased, and they are better adapted to improve their stability [156,157]. In the same way, bacteria, especially lactic acid bacteria, are widely selected for use in industrial bread making [71].

Many valuable properties observed in the select starters of bacteria and yeasts are related to their enzyme activities or involvement in the enzyme activities in the sourdough system. The wheat grain has enzymes that can be activated or inhibited by the low pH of the sourdough environment, and the microbiota present in sourdough contributes to its enzymes in the sourdough metabolic process [3]. The mechanisms of LAB and yeast enzymatic action during sourdough fermentation are becoming well-known [54]. The hydrolases demonstrate an essential role in sourdough bread. However, other enzyme classes, including transferases and oxidoreductase, are involved in the properties of sourdough fermentation [81,158]. Figure 3 shows the sourdough fermentation process and the major enzymes involved.

### 5.1. Transferases

The more expressive metabolic activities of microbial communities during fermentation are acidification, leavening, and flavor formation, all related to the metabolism of carbohydrates [159]. Transferases are enzymes that catalyze the transfer of specific functional groups from one molecule to another. Glycosyltransferases (EC 2.4) are involved in the biosynthesis of exopolysaccharides in sourdough. It is an enzymatic group produced by the LAB from sourdough microbiota, such as *Lactobacillus*, *Leuconostoc* spp., *Leuconostoc citreum*, and *Weissella* species. Glucansucrases are glycosyltransferases from LAB that split sucrose. The resulting glucose builds EPS-type α-glucan polymers such as dextran, mutan, alternan, and reuteran. Fructansucrase, found in Gram-negative and Gram-positive bacteria, is another glycosyltransferase that synthesizes levan (levansucrase) or inulin (inulosucrase). The inulosucrase is observed only in LAB [160,161].

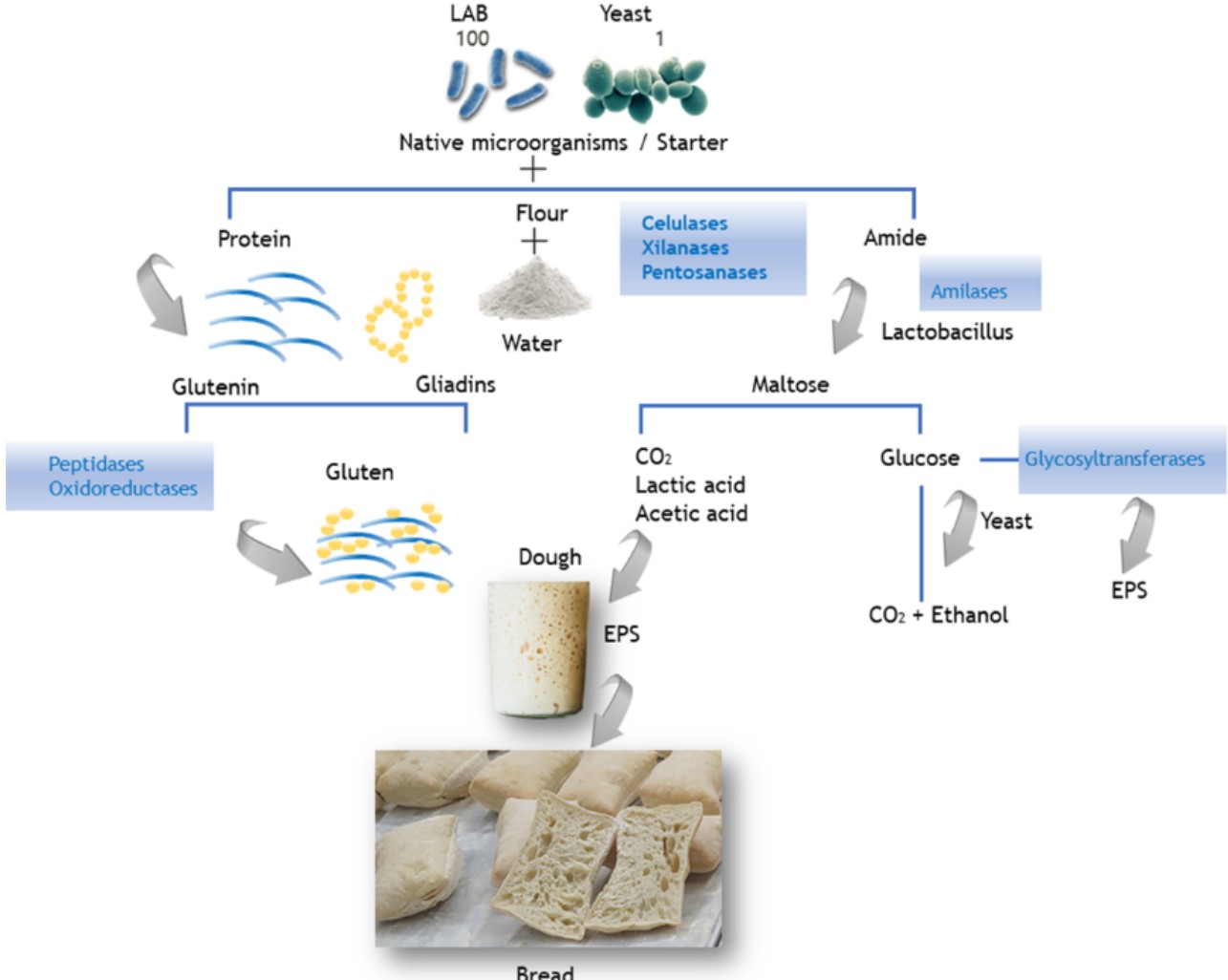

**Figure 3.** General scheme of bread sourdough fermentation and the major enzymes involved. EPS: Exopolysaccharides.

EPS formed during sourdough fermentation influences the dough's viscoelastic properties and improves the texture and shelf-life. In this context, EPS can replace hydrocolloids used as bread improvers and meet consumer demands for reduced use of food additives [162].

Adding a glucosyltransferase B, made by *Streptococcus thermophilus* with the activity of 4,6-α-glucanotransferase in the preparation of wheat flour dough, results in bread with better health benefits. This enzyme increased the number of short branches and percentage of (α1→6) linkages in starch, resulting in slow starch digestibility and low retrogradation properties. It also reduced hardness, gumminess, and chewiness, indicating improvements in texture [92].

Transglutaminase (TGase, EC 2.3.2.13) catalyzes the cross-linking of proteins between the ε-amino group of a lysine residue and the γ-carboxamide (acyl) group of a glutamine residue. Which results in an intra- or intermolecular bridge highly resistant to proteolysis, which leads to an increase in the structure and texture of protein substrates [163,164]. It is an important enzyme for the food industry, where studies report that its application improves dough handling properties, increases bread stability, and volume. The protein polymers resulting from the action of TGase can modify the rheological properties of gluten, transforming a weak gluten into a strong one [75].

### 5.2. Oxidoreductases

Oxidative enzymes are applied to treat wheat flour to restore the gluten network. Those enzymes can be divided into exogenous oxidative enzymes (laccase, glucose oxidase, hexose oxidase) and endogenous oxidative enzymes (tyrosinase, peroxidase, catalase, sulfhydryl oxidase, lipoxygenase, protein disulfide isomerase, NAD(P)H-dependent dehydrogenase, thioredoxin reductase, and glutathione reductase) [165].

Oxidoreductase (EC 1) catalyzes reactions of electron transfer. Glutathione reductase (EC 1.8.1.7) is the most known enzyme in this group related to sourdough bread properties. It acts on a sulfur group of donors, with NAD+ or NADP+ as the acceptor [166].

Gluten protein presents intra- and intermolecular chains of disulfide bonds between amino acids [167]. The glutathione reductase activity, identified in the microbiota of natural fermentation, interferes with these thiol bonds [158]. The glutathione reductase deficient mutant strain (*Lactobacillus sanfranciscensis*) produced bread with a softer texture and higher specific volume than bread made with traditional biological yeast and non-mutant *Lactobacillus sanfranciscensis* activity [167].

Glucose oxidase (EC 1.1.3.4) and pyranose oxidase (EC 1.1.3.10) are oxidoreductases produced by several fungi, mostly from *Aspergillus niger* in the case of glucose oxidase. One molecule of glucose oxidase catalyzes the conversion of β-D- glucose into hydrogen peroxide ($H_2O_2$) and gluconic acid [168]. According to Xu et al. [169] the resulting $H_2O_2$ can further oxidize the free sulfhydryl units in the gluten protein and promote the formation of disulfide bonds in the gluten network, thereby strengthening the gluten network. In addition, glucose oxidase has antimicrobial activity against fungi and foodborne bacteria [170]. These effects enhance the rheological properties, volume and height, and stability of the bread [169,171,172].

Laccases (benzenediol: oxidoreductase oxygen, EC 1.10.3.2) are multi-copper enzymes and they belong to a group of so-called polyphenol oxidases. They catalyze the oxidation of a wide range of substrates while simultaneously reducing molecular oxygen to water [173–175]. Laccase activity has been found to enhance the viscoelastic dough qualities and in recent years it has been used as a dough and bread improver, primarily utilized in commercial preparations [175,176]. Pentosans are polymerized by laccase and favor the aggregation of glutenins, which lowers the protein content of glutenin macropolymers. While not all researchers agree, some authors report that laccases are also important in baking because they can cross-link the arabinoxylans found in whole flour, improving the gluten index and inducing hetero-crosslinking between arabinoxylans and the gluten matrix [165].

### 5.3. Lyases

Lyases (EC 4) are a group of enzymes that cleave carbon bonds with oxygen, carbon, nitrogen, and others and eventually form double bonds or a new ring [177].

The glutamate decarboxylase (EC 4.1.1.15) is commonly found in the sourdough environment and acts in L-glutamate's decarboxylation to form γ-aminobutyric acid (GABA), a non-protein amino acid with various health benefits.

Amino acid catabolism by LAB is related to sourdough properties such as taste and nutrition. However, these reactions can accumulate toxic biogenic amines, especially tyramine, cadaverine, and putrescine. It is well-known that a high GABA content in foods is highly desirable as it is a non-protein amino acid with many health benefits, and some LAB, such as *Levilactobacillus brevis*, are excellent producers [178,179]. LAB and yeast are the most important GABA producers because they are commercially valuable as starters in fermented foods. GABA has several physiological functions, such as inhibiting neurotransmission in the central nervous system. Other related functions include relaxation, sleeplessness, enhanced immunity under stress, increasing the concentration of growth hormone in plasma, preventing diabetes, inhibiting the invasion and metastasis of various types of cancer, anti-inflammatory effects, blood pressure regulation, and antioxidant effects. It is also a hormonal regulator (for review, see Diana et al., 2014 [178] and Polak et al. 2021 [179].

Indeed, fermented foods represent an excellent source of dietary GABA. In a recent study, *Lactobacillus brevis* A7 and *Lactobacillus farciminis* A11, demonstrated high GABA-producing capability when used to prepare bread containing 20% amaranth flour (amaranth has a high concentration of essential amino acids) [180].

Calabrese et al. [181] analyzed spontaneous sourdough metagenomes and transcriptomes to identify microorganisms and their role in the fermentation process. Based on meta-omics data collected and their identified microbiota, a reconstructed synthetic microbial community was established. The results revealed a complex network with dominant, subdominant, and satellite communities engaged in different functional pathways. Aspartate ammonia-lyase (EC 4.3.1.1), the enzyme that converts aspartate into fumarate, was uniquely transcribed by *Lactiplantibacillus plantarum*. This approach was effective and significant for studying these complex metabolic frameworks bringing new knowledge into this complex fermentation.

*5.4. Hydrolases*

Hydrolases (EC 3) are enzymes that use water to break various chemical bond types.

Amylase is the most popular hydrolase in baked good processes. However, peptidases have received more attention recently due to investigations into their role with gluten. Other hydrolases have essential roles in the baking processes, such as cellulase, xylanase, and pentosanase [182]. Table 4 presents the main enzymes used in the bakery and the microorganisms that produce them, according to Dahiya et al. [183].

**Table 4.** List of microorganisms that produce the main enzymes used in the bakery.

| Enzyme | Microorganisms | Reference |
|---|---|---|
| Xylanase | *Sporotrichum thermophile* BJAMDU5 | [184] |
| | *Pichia pastoris* | [185] |
| | *Bacillus subtilis* | [186] |
| | *Myceliophthora thermophila* BJTLRMDU3 | [187] |
| Phytase | *Lactobacillus casei* | [188] |
| | *Enterobacter* sp. ACSS | [189] |
| | *Sporotrichum thermophile* | [190] |
| | *Aspergillus niger* NCIM 563 | [191] |
| Amylase | *Rhizopus oryzae* | [192] |
| | *Bacillus subtilis* US586 | [193] |
| | *Bacillus subtilis* M13 | [194] |
| | *Streptomyces badiun* DB-1 | [195] |
| Glucose Oxidase | *Aspergillus niger* | [196] |
| | *Penicillium notatum* | [197] |
| | *Aspergillus niger* | [198] |
| | *Aspergillus niger* | [199] |
| Peptidase | *Rhizopus oryzae* | [192] |
| | *Bacillus subtilis* PF1 | [200] |
| | *Bacillus subtilis* | [201] |
| | *Bacillus pumilus* SG2 | [202] |
| Lipase | *Aspergillus niger* MTCC 872 | [203] |
| | *Pseudomonas fluorescens* (NRLL B-2641) | [204] |
| | *Bacillus subtilis* I-4 | [205] |
| | *Bacillus* sp. MPTK 912 | [206] |
| Cellulase | *Sporotrichum thermophile* BJAMDU5 | [203] |
| | *Streptomyces strain* C188 | [207] |
| | *Cellulomonas uda* | [208] |
| | *Trichoderma reesei* NCIM 1186 | [209] |

5.4.1. Amylase, Inulinase, and Their Impact on Bread Structure and Properties

Several types of amylases act on starch, mainly liberating glucose for the fermentation and improvement of the bread's color, flavor, and shelf-life [210]. The main groups are endoamylases, exoamylases, and debranching amylases. Among the endoamylases, $\alpha$-

amylase (EC 3.2.1.1) is the main one, acting on α-1,4 bonds in the amylose and amylopectin chains. The group of exoamylases includes glucoamylase (EC 3.2.1.3), α-glucosidase (EC 3.2.1.20), and β-amylase (EC 3.2.1.2) promoting the hydrolysis of terminal glycosidic units. Debranching amylases act on α-1,6 branch bonds, and pullulanase (EC 3.2.1.41) is the leading representative [211].

Starch/amide is present in around 70–75% of wheat flour. It comprises a linear chain of α-1,4 glycopyranoside (amylose) and a linear chain of α-1,4 glycopyranoside with α-1,6 branch points (amylopectin). During the baking process, the bread undergoes several changes in starch retrogradation relative to the crystalline structure of amylose and amylopectin. These changes occur not only in baking but also during the cooling and storage processes [212]. Starch retrogradation is a process in which disaggregated amylose and amylopectin chains in a gelatinized starch paste reassociate to form a more ordered structure [213]. It is a pivotal issue in bread aging and is the cause of bread hardening.

The amylases produced by LAB strains reduce the aging process during bread storage [214]. Amylase-producing microorganisms in the sourdough environment are essential in converting starch into fermentable carbohydrates such as maltodextrins, maltose, sucrose, and glucose. LAB are sources of maltose phosphorylase, which generate D-glucose, β-D-glucose, 1-phosphate, and 1,6-α-glucosidase, which hydrolyzes α-(1–6)-glucooligosaccharides [3]. In addition, wheat has α-amylase, β-amylase, and glucoamylase activity, but at pH < 4.5, only glucoamylase maintains its activity. Microorganisms in the sourdough microbiota produce inulinase (β-2, 1-D-fructan-fructan-hydrolase—EC 3.2.1.7)—a glycosylase that hydrolyzes the β-2.1 bonds of fructose of the fructose polymer, inulin. Inulinase can reduce the number of oligosaccharides, disaccharides, monosaccharides, and fermentable polyols (FODMAPs) in bread [78]. FODMAPs are associated with irritable bowel syndrome. However, it is not appropriate to eliminate these polyols but only to reduce them to an adequate level, as these compounds are considered prebiotics to maintain the healthy intestinal microbiota [215]. *Kluyveromyces marxianus* yeast strains have reduced the concentration of FODMAPs in whole wheat bread [78].

### 5.4.2. Cellulase, Phytase, and Xylanase for Mineral Bio-Accessibility Improvement in Bread

Cellulases are classified into endoglucanases (act in the middle of cellulose) and exoglucanases (cleave the extremities of the polymer). The endoglucanase is the endo-β-1,4-glucanase (EC 3.2.1.4), cleaving intramolecular bonds of β-1,4-glycosidic. The exoglucanases are the β-glucosidase (EC 3.2.1.21) and exo-β-1,4-glucan cellobiohydrolase (EC 3.2.1.91), which cleave ends of the cellulose and glycosidic terminals (liberating Cellobiose), respectively [216].

Endocellulase acts on cellulose and β-glucan substrates, removing insoluble arabinoxylans, contributing to gluten network formation [217]. Subsequently, there is a decrease in the hardness of the bread and, consequently, an improvement in the sensory evaluations. Cellulase action may also provide an anti-staling effect, which may be related to alterations in water distribution between the starch–protein matrix [182]. In the study carried out by Li et al. (2014) [218], the authors added β-glucanase to a mixture of barley flour (30%) and wheat flour (70%) with approximately 1.5% glucan as the starting material. As a result, they noticed an improvement in the maneuverability of the dough, an increase in softness and elasticity, an increase in the specific volume of the bread, and a reduction in hardness.

The enzyme phytase or myoinositol-hexaphosphate phosphohydrolase is present in plants, bacteria, yeasts, and filamentous fungi. It hydrolyzes phytate molecules and releases inorganic phosphate, dephosphorylation-dependent inositol esters, and minerals linked in the phytate structure [219]. Two classes of phytase are categorized depending on where the hydrolysis of phytate begins, -3-phytase (EC 3.1.3.8) or 6-phytase (EC 3.1.3.26); the 3-phytase being the one that removes the orthophosphate group from the position C3 characteristic of microorganisms. Two classes of phytase are found depending on the position where the hydrolysis of phytase -3-phytase (EC 3.1.3.8) or -6-phytase (EC 3.1.3.26).

The 3-phytase removes the orthophosphate group from the position C3 characteristic of microorganisms [220].

Xylan is the main hemicellulosic component of hardwoods and accounts for approximately 30% of the woody cell wall. Xylanases (EC 3.2.1.8) are enzymes that hydrolyze this polysaccharide. They are of two types: β-1,4-endoxylanase cleaving internal glycosidic linkages, and α-D-xylosidase which cleaves xylo-oligosaccharides, and xylobiose forming xylose, a disaccharide of two xylose molecules [211]. It is important to highlight that xylanase from various sources has different mechanisms. Two mechanisms can summarize the improvement in bread quality due to the addition of xylanase: (i) the removal of arabinoxylans from gluten alters the distribution of water between gluten proteins and arabinoxylans; (ii) pentosan destruction and viscosity reduction effect [182].

Aslam et al. [221] reviewed that wheat is a reservoir of minerals, mainly in the aleurone tissue between the endosperm and the seed coat. However, these minerals are complexed with phytate molecules that reduce their bio-accessibility. Xylanase and cellulase showed satisfactory results to facilitate access into the aleurone tissue for the action of phytase, improving iron bio-accessibility [222].

Although some aspects of mineral assimilation by the human body need to be considered, such as diet, blood flow, gut epithelium, and intestinal microbiota [223], some LAB and yeasts may have phytase activity that improves mineral bio-accessibility in sourdough bread [79,224–226]. *Bifidobacterium* strains have demonstrated the ability to increase the accessibility of iron through its phytase activity during bread fermentation [107].

Pentosanase is a hemicellulase that acts on pentosans, particularly hydrolyzing water-unextractable arabinoxylan (WUA) into water-extractable arabinoxylan (WEA). The WUA represents 85% of the total arabinoxylan in whole wheat flour. The WEA affects the bread making process by reducing viscosity and making the dough more malleable. However, WUA promotes lower bread quality for interfering with the gluten network because it competes with gluten for water [169]. Adding pentosanase has shown a good outcome for bread making, with higher specific volume and crumb texture improvement [227]. Moreover, Martínez-Anaya et al. [228] showed that sourdough with starter modified the results of pentosanase used in the bread making process, with the fraction of soluble pentosan increased considerably, and a lower xylose/arabinose ratio.

### 5.4.3. Lipase and the Baking Technology

Lipase and esterase (EC 3.1) catalyze the reaction that forms ester bonds. The subclass triacylglycerol acyl hydrolase (EC 3.1.1.3) acts on triglycerides releasing fatty acids and glycerol.

Both phospholipase A1 (EC 3.1.1.32) and phospholipase A2 (EC 3.1.1.4) liberated fatty acids on phospholipids and other lipophilic compounds. These groups are the most important in the bread making process [210,211,229].

In baking, using lipase to replace emulsifiers, such as monoglycerides, proved to be an efficient alternative to replace chemical additives [230]. Triacylglycerol acyl hydrolase can break down triglycerides, which interferes negatively with the gas retention gluten properties [210]. Sourdough fermentation inhibits the lipase activity due to decreased pH in the system, which is an efficient alternative for neutralizing the lipase activity in wheat germ [231]. Wheat germ, rich in vitamins, amino acids, fiber, and minerals, is nutritionally more important in wheat. However, it causes oxidation in wheat flour during storage because this needs to be removed before producing wheat flour [232]. A study showed that wheat germ could produce bread with a superior nutritional, textural, and sensory quality after the sourdough process to obtain sourdough-fermented wheat germ free of oxidation potential [233].

### 5.4.4. Peptidase and Implications for the Gluten Network

Peptidases hydrolyze peptide bonds of proteins and peptides. They are still known as proteases, proteinases, and proteolytic enzymes [234].

The International Union of Biochemistry and Molecular Biology (IUBMB) classifies peptidases with criteria based on reaction and catalytic type. Endopeptidases (EC 3.4.21-99) hydrolyze internal peptide bonds, while exopeptidases (EC 3.4.11-19) remove amino acids from the ends of peptide chains. These enzymes have a complex classification involving the reaction and catalytic type and the evolutive relations. The MEROPS platform is a complete site about peptidases, including their complete classification [234].

Gluten is a complex protein responsible for wheat bread's most peculiar characteristics, such as expandability and retention of gas produced in fermentation. Its quality is more important than quantity [235]. Gluten structure varies with the cultivar and cultivation conditions [236]. It is composed of gliadin (α, β, γ, ω), glutenin, and polypeptide fractions that combine with gliadin or glutenin [237]. Figure 4 seeks to demonstrate the complexity of gluten formation and the water effect. Gluten forms a network retaining carbon dioxide during baking, making dough expand. It is responsible for dough's rheological characteristics due to gliadin, which ensures the dough's viscosity, and glutenin, with the property of elasticity. During sourdough fermentation, the partial hydrolysis of gliadin and glutenin proteins occurs because of the acidification and activation of cereal peptidases. In addition, endogenous flour peptidases became activated at a low pH reducing the gluten disulfide bonds [38]. Cysteine proteinases in wheat grain are capable of hydrolyzing both gliadins and glutenin. The hydrolysis of glutenins during sourdough fermentation results in depolymerization and subsequent solubilization [238]. Endopeptidases may be applied to improve the rheological properties, and one example is the endo- and exopeptidases from *A. oryzae*. The fungi peptidases are employed to modify wheat gluten by limited proteolysis resulting in better rheology and shorter dough mixing for a higher loaf [239]. Lactic acid bacteria also exhibit proteolytic activity, decreasing the content of native gluten [231,240,241]. Several studies have focused on LAB peptidases and demonstrated that in different ways, that LAB in sourdough fermentation has a proteolytic activity for gluten hydrolysis [60]. Celiac disease or gluten enteropathy is an autoimmune disease caused by gluten-containing food and results in those with it to adhere to a strict gluten-free diet. One characteristic of gluten is that it is resistant to proteolysis by human digestive peptidases [242]. There is considerable interest in producing gluten-free bread that is specific for celiac consumers. The gliadin fraction, especially α2-gliadin, has been considered the most allergenic part of gluten [243]. Another possible treatment is enzyme therapy, in which peptidases are ingested with food to increase the hydrolysis of resistant immunogenic peptides containing proline and glutamine amino acids [244]. The hydrolysis of some parts of gluten could favor the appearance of allergen terminals and cause more harm than benefit to a celiac person [243]. On the other hand, hydrolyzing gluten to reduce the allergenic components potentially benefits non-celiac people [69]. A group of LAB hydrolyzed gluten, but not the most allergenic part in the central region of alpha-gliadin. However, other allergenic proteins were degraded, improving the wheat digestibility [245]. A bioprospection analysis of gliadin-cleaving proteolytic activity among 20 *Lactobacillus* strains showed that the most active strain, *L. casei*, was able to hydrolyze the 33-mer immunogenic peptide of α-gliadin by 82% in 8 h and completely in 12 h [246]. Bread digestibility might be positively affected by long sourdough fermentation [148]. *L. rhamnosus, Pediococcus pentosaceus*, and *Lactobacillus curvatus* used gluten as the only source of nitrogen, significantly reducing the allergenic composition of wheat and improving digestibility [69]. Strains of *Lactobacillus plantarum* ES137 and *Pediococcus acidilactici* ES22, isolated from sourdough fermentation, showed high proteolytic capacity [247]. Lactobacilli in sourdough have proteolytic capacity to produce γ-glutamyl dipeptides, other peptides, and amino acids as flavor precursor compounds in the bread produced [248]. Other bacteria, such as *Bacillus* spp. isolated from sourdough, are capable of gluten hydrolysis [140].

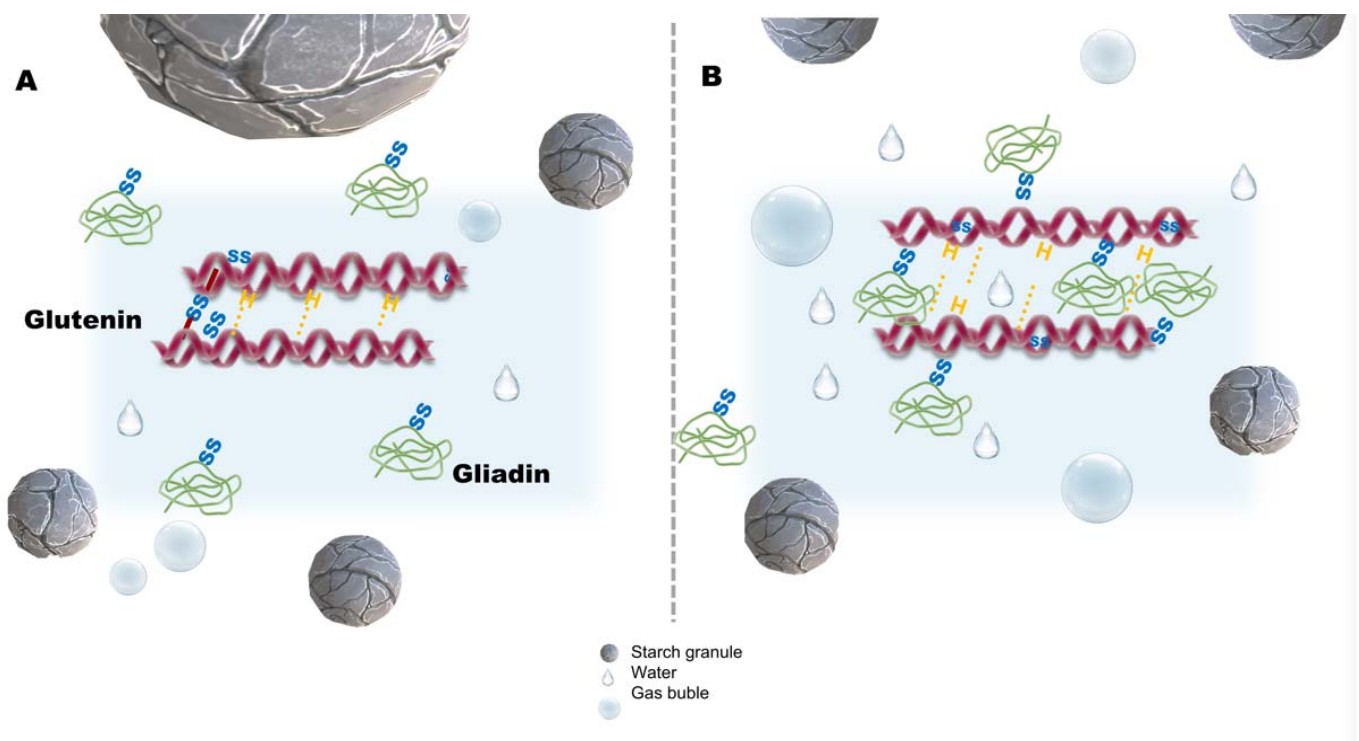

**Figure 4.** Some possible interactions of wheat proteins. (**A**)-low water content. (**B**)-baking mixture with optimal water content, based on the work of Feng et al. and Schopf and Scherf [249,250].

## 6. General Regulation for Microbes Used in Sourdough Bread

Searching for rules that maintain food safety is a critical concern due to the growing use of microorganisms as foods or probiotics. At this moment, there is an effort at Codex Alimentarius to harmonize rules for probiotics and the use of microorganisms in food [251].

In the United States, there are rules defined by the Food and Drug Administration (FDA) for the use of microorganisms in food production. It is necessary to qualify as a Generally Recognized as Safe for Use (GRAS) microorganism [252]. In Brazil, the biological fermenting mixtures and their microbial content are classified as a supporting and processing agent and are exempted from registration by ANVISA [253]. However, food with nutritional allegations or probiotic use needs to be registered [253], and microorganisms isolated from national raw materials, or related territory, are recognized as national genetic patrimony [254], and they need to be registered too.

The requirements for a microorganism to be considered GRAS include taxonomic identification, absence of virulence, enterotoxins, hemolytic activity, and transferable antibiotic resistance genes. A probiotic microorganism still needs to prove stress tolerance (low pH, gastric enzymes, bile salts) up to levels found in the intestine, as well as adhesion capacity and antipathogenic activity, and to be approved in clinical trials [255]. Moreover, FAO/WHO advises in its guidelines that the evaluation of probiotics should be done using molecular information from strains, because the probiotic properties are related to this level of identification [256].

In France, since 1993, a law specifies the characteristics of sourdough bread, such as pH 4.3, acetic acid 900 ppm, at least, with sourdough consisting of lactic acid bacteria and yeasts. It is possible to add *Saccharomyces cerevisiae* to a maximum of 0.2% on the flour weight. If sourdough is dry, it needs at least one billion live food bacteria and one to ten million live yeasts per gram [257].

In Europe, since 1997, with the novel food regulation, all food that started to be produced after 15 May 1997 is considered a novel food and has specific rules to be followed [258]. For microorganisms usually found in sourdough and used as a starter, it is

challenging to define it as a novel food because sourdough predates 1997 [259]. Therefore, if the starter used in the sourdough fermentation is unusual or has some additional novel properties, it could be considered a novel food. For instance, UV-treated baker's yeast (*Saccharomyces cerevisiae*) has been considered a novel food since 2017 because it is alleged to increase vitamin D2 [260]. Moreover, the EU1151/2012 recognizes that baker products could have a geographical denomination [11].

Regarding microorganism patents, the United States, Japan, Europe, India, and participants of The Trade-related Aspects of Intellectual Property Rights (TRIPS) agreement consider that microorganisms should be patented [261]. However, Brazil, for instance, allows patenting only for genetically modified microorganisms [262].

## 7. Conclusions

The role of microorganisms in sourdough bread has been extensively demonstrated in the literature. Increasing information is arising about the fermentation process and other metabolic routes promoting improvements in quality and adding interesting nutritional, health, and sensory characteristics. Currently, several benefits of probiotic and postbiotics elements are well established in terms of the bread's properties, such as rheological and organoleptic properties, as well as human health and dough quality. However, there are challenges to overcome to produce a microbial starter or consortium with high technological properties on an industrial scale. Better digestibility, satiety, and antioxidants, among other properties, could be obtained. The sourdough ecosystem needs to be explored more concerning the dynamic of the interaction among the microorganisms, including the quorum sensing process and the metabolic regulation of sourdough. Molecular and omics tools are bringing more information about the sourdough microbiome and the complex metabolic network, allowing the complete identification of the non-cultivable microbial population and establishing probiotic strains. A better characterization of the postbiotics present in the dough is also essential. Another promising area of study is relative to the addition of other microbial products that can improve the final quality of bread. Sourdough bread and the microorganisms involved in its production are an area for increased research activity. The breaking and sharing of bread is of biblical and global importance. Further studies will elucidate the mechanisms of action of these bioproducts, metabolic routes, and their effects on bread, and the production of safe food leading to technological advances.

**Author Contributions:** Conceptualization and writing, I.T.A.; writing, formal analysis, original draft preparation, F.R.P.M.; conceptualization, writing, review, editing, project administration, and funding acquisition, A.B.V. All authors have read and agreed to the published version of the manuscript.

**Funding:** This research was funded in part by the postgraduate program of the Paulo de Góes Institute of Microbiology, Federal University of Rio de Janeiro (UFRJ), through the Coordenação de Aperfeiçoamento Pessoal de Nível Superior (CAPES) [grant number 001], Conselho Nacional de Desenvolvimento Científico e Tecnológico (MCTI-CNPq) grant code [309461/2019-7], and Fundação de Amparo à Pesquisa do Estado do Rio de Janeiro (FAPERJ), grant code ["CIENTISTA DO NOSSO ESTADO" 26/202.630/2019 247088].

**Institutional Review Board Statement:** Not applicable.

**Informed Consent Statement:** Not applicable.

**Data Availability Statement:** Not applicable.

**Acknowledgments:** We would like to thank Andrew Macrae for the English review and Slide-Model.com for the diagram in Figure 1. The images in the graphical abstract are from TogoTV (©2016 DBCLS TogoTV) and Servier Medical Art (smart.servier.com).

**Conflicts of Interest:** The funders had no role in the design of the study; in the collection, analyses, or interpretation of data; in the writing of the manuscript; or in the decision to publish the results.

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
