# Peer review of "Probiotics in the Sourdough Bread Fermentation: Current Status"

_fermentation, doi:10.3390/fermentation9020090_

Round 1
Reviewer 1 Report
In this study, the current status of the probiotics in the sourdough bread fermentation were summarized. This review focuses on the main steps of sourdough fermentation, the microorganisms involved, and advances in bread production with functional properties. The impact of probiotics on human health, the metabolites produced, and the main microbial enzymes used in the bakery industry were also discussed. The written was relatively simple and unsatisfied. Some specific comments are also below:
1. The overall analysis of the manuscript is superficial, and it is suggested that part of the analysis is more in-depth, in order to reflect the depth of the analysis of the paper.
2. In line 132-134, the authors stated that “The authors propose probiotics to different yeast groups in sourdough: : the first group, Pichia kudriazevii and or Torulaspora delbrueckii, prevail; the second group, Wickerhamomyces anomalus and or Candida glabrata, prevail; and the third group, Saccharomyces cerevisiae”. Please make further analysis, and give the reasons for your suggestion.
3. Please adjust the Table 1.
4. In line 142-143, the predictive model is missing, and please supply the predictive model.
5. The authors pointed out that the glucosyltransferase B produced by LAB resulted in low digestibility of starch. Does it mean that excessive LAB quantity should be controlled?
6. In line 303-304, please supply the functional role of GABA on the sourdough.
7. In line 318-321, please add the function of the listed enzymes in the sourdough instead of just pentosanase.
8. In line 431-432, what are the microorganisms in sourdough fermentation and how to play proteolytic activity for gluten hydrolysis?
Author Response
In this study, the current status of the probiotics in the sourdough bread fermentation were summarized. This review focuses on the main steps of sourdough fermentation, the microorganisms involved, and advances in bread production with functional properties. The impact of probiotics on human health, the metabolites produced, and the main microbial enzymes used in the bakery industry were also discussed. The written was relatively simple and unsatisfied. Some specific comments are also below:
1.The overall analysis of the manuscript is superficial, and it is suggested that part of the analysis is more in-depth, in order to reflect the depth of the analysis of the paper.
The sections and the content of the review were modified and discussed more in-depth, as requested by the reviewer.
2.In line 132-134, the authors stated that "The authors propose probiotics to different yeast groups in sourdough: : the first group, Pichia kudriazevii and or Torulaspora delbrueckii, prevail; the second group, Wickerhamomyces anomalus and or Candida glabrata, prevail; and the third group, Saccharomyces cerevisiae". Please make further analysis and give the reasons for your suggestion.
It is not our suggestion. We cited the article of Kerrebroeck et al. (2017). The authors did a meta-analysis to study species diversity (Lab and yeasts) and process conditions (Type I and II). to improve the understanding, we replaced the text as described below.
“A meta-analysis with 583 sourdough-related literature articles in the period 1999-2017 was done by Kerrebroeck et al. 2017 [34] based on the study of species diversity (LAB and yeasts) and process condition (Type I and II). The authors concluded that the number of yeast is lower than LABs and that yeast microbial diversity of type I Sourdoughs is lower than type II. Through the Principal component (PC) analysis, three clusters were detected. The first group has a high relative abundance of Pichia kudriavzevii and/or Torulaspora delbrueckii. In the second group, a high relative abundance of Wickerhamomyces anomalus and/or Candida glabrata was established, and the third Saccharomyces cerevisiae and/or C. humilis species were detected.”
- Please adjust the Table 1.
As requested by the reviewers, the table was replaced by a figure.
- In lines 142-143, the predictive model is missing, and please supply the predictive model.
As suggested by the reviewer, the text as replaced by:
“The growth kinetics of six strains of LAB: Fructilactobacillus sanfranciscensis, were compared with five strains of yeast: Kazachstania humilis, from sourdough, by Altilia et al. 2021 [29]. Three predictive models were used to evaluate the behavior of co-cultivated microorganisms: The zwitweringg model based on Gompertz's equation, Baranyil and Roberts' function, and Schiraldi's function. The results showed that F. sanfranciscensis strains significantly steer the growth kinetics affecting the ratio of bacterial to yeast cells, the yeast strains of K. humilis adapt to the bacterial strains. The authors discuss the possibility of metabolic interactions for stabilizing the sourdough consortium through communication, like quorum sensing, to control population density, among other functionalities. In another co-culture article with LAB and yeast, the yeast population size decreased in the presence of LAB regardless of the strain, while the LAB´s population size was rarely influenced by the presence of yeast [30].”
- The authors pointed out that the glucosyltransferase B produced by LAB resulted in low digestibility of starch. Does it mean that excessive LAB quantity should be controlled?
The foodborne lactic acid Streptococcus thermophilus produced glucosyltransferase. This enzyme was added to wheat flour dough to improve the texture and health benefits. It is an experimental study with the potential to be applied in the bakery after more consolidated experiments. Several factors influence the ecosystem of sourdough. Changing one isolated factor may not produce the desired effect in this context.
The text was changed as described below :
“Adding a glucosyltransferase B, made by Streptococcus thermophilus with the activity of 4,6-α-glucanotransferase in the preparation of wheat flour dough, results in bread with better health benefits. This enzyme increased the number of short branches and percentage of (α1→6) linkages in starch, resulting in slow starch digestibility and low retrogradation properties. It also reduced hardness, gumminess, and chewiness, indicating improvements in texture [99].
- In lines 303-304, please supply the functional role of GABA on the sourdough.
There are studies relating the importance of GABA in sourdough, but the functions described are the same for other fermented foods; however, we improved the explanation, and the text was replaced.
“The glutamate decarboxylase (EC 4.1.1.15) is commonly found in the sourdough environment and acts in L-glutamate's decarboxylation to form γ-aminobutyric acid (GABA), a non-protein amino acid with various health benefits. Amino acid catabolism by LAB is related to sourdough properties such as taste and nutrition. However, these reactions can accumulate toxic biogenic amines, especially tyramine, cadaverine, and putrescine. It is well known that a high GABA content in foods is highly desirable as it is a non-protein amino acid with many health benefits; and some LAB, such as Levilactobacillus brevis, are excellent producers [189,190]. LAB and yeast are the most important GABA producers because they are commercially valuable as starters in fermented foods. GABA has several physiological functions, such as inhibiting neurotransmission in the central nervous system. Other related functions include relaxation, sleeplessness, enhanced immunity under stress, increasing the concentration of growth hormone in plasma, preventing diabetes, inhibiting the invasion and metastasis of various types of cancer, anti-inflammatory effect, blood pressure regulator, and antioxidant effect. It is also a hormonal regulator (for review, see Diana et al., 2014 [189] and Polak et al. 2021 [190]. Indeed, fermented foods represent an excellent source of dietary GABA. In a recent study, Lactobacillus brevis A7 and Lactobacillus farciminis A11, demonstrated high GABA-producing capability when used to prepare bread containing 20% amaranth flour (Amaranth has a high concentration of essential amino acids) [191].”
- In lines 318-321, please add the function of the listed enzymes in the sourdough instead of just pentosanase.
The correction was done as described below.
“Pentosanase is a hemicellulase that acts on pentosans. Particularly hydrolyzing water-unextractable arabinoxylan (WUA) into water-extractable arabinoxylan (WEA). The WUA represents 85% of the total arabinoxylan in whole wheat flour. The WEA affects the breadmaking process by reducing viscosity and making the dough more malleable. However, WUA promotes lower bread quality for interfering with the gluten network because it competes with gluten for water [180]. Adding pentosanase has shown a good outcome for breadmaking, with higher specific volume and crumb texture improvement [239]. Moreover, Martínez-Anaya et al. [240] showed that sourdough with starter modified the results of pentosanase used in the breadmaking process, with the fraction of soluble pentosan increased considerably and a lower xylose/arabinose ratio.”
- In line 431-432, what are the microorganisms in sourdough fermentation, and how to play proteolytic activity for gluten hydrolysis?
This information has been added to the text.
Reviewer 2 Report
In the reviewed manuscript, Akamine et al. elaborate on the use of probiotics in sourdough bread production. The article begins with a brief introduction, covering three short paraphs. Here, the importance of sourdough fermentation and the positive effects of probiotics are described. Finally, the authors state the aim of the article, which is “…to review the improvement of knowledge to use sourdough fermentation, its enzymes, metabolites, and technology advance to produce sourdough bread with desirable properties.“ After the introduction, the article continues with a second chapter, giving an overview of common sourdough fermentation processes. Then, the article continues with a large third chapter, in which the sourdough microbiota is addressed. Here, one subchapter (3.1.) also addresses probiotics accordingly. Next, chapter four deals with enzymes' role in sourdough fermentation. The review is complemented by a fifth chapter, discussing regulatory issues concerning the use of microbe addition during sourdough bread production. Finally, the article closes with a short conclusion, stating that probiotics present in sourdough fermentation, as a fact, “promote improvements in quality, adding interesting nutritional, health, and sensorial characteristics, especially in bread.“ But simultaneously, further studies have to be conducted to improve the understanding of microbes by omics technologies and synthetic biology, and also, it has to be ensured that the final food product is safe to use for the customers.
In general, the topic is very interesting and no other review article could be found that mainly addresses the subject of probiotics in sourdough fermentation. Nonetheless, the review should not be accepted to Fermentation (reject) in its current quality. However, I hope my comments below are helpful and will contribute to improving the quality of the manuscript.
General comments:
1. The work's title suggests that the manuscript clearly focuses on probiotics in sourdough bread fermentation and gives an overview of the current research status on this topic. However, after reading the manuscript, I am not convinced that the exact effect of probiotics is the work's focus. Instead, it seems to be a more general review of sourdough fermentation. It is, of course, necessary to give information on related areas, but it would have been much appreciated if these were more concise and probiotics treated to more extent and with a more sophisticated focus.
2. Reading the article, I wondered why the authors had decided to structure it in its current form. Focusing on probiotics, it would have maybe been better to choose this as a single and major chapter and derive all other content and required background knowledge with current references from this. The present structure makes it hard to follow the article's key message. Also, although often containing interesting information, some parts of the manuscript do not appear to be well connected to previous parts – leading to some hard interruptions for the reader when working through the manuscript. For a review article, this should be improved.
3. The quality of the manuscript is not sufficient – this does not only concern language but also some minor careless mistakes, which are not only a few. A manuscript submitted for peer review should be of higher quality. I hope my specific comments will also be helpful in this context.
Specific comments:
1. Line 15: The abstract mainly uses the present tense; therefore, change “were” to “are” to keep it consistent.
2. Line 22: Remove the comma before the reference “..,[1-3]”.
3. Line 28: Change “sensory food” to “food sensory”.
4. Lines 28-30: “It is possible to find sourdough in the bakery industry, in liquid, in paste, or dry form with living microorganisms through the development of technologies, like spray drying, fluidized bed, or freeze-drying.” The message of this sentence is not clear to me. The named drying technologies have been well-developed for decades, maybe “application” would be better in this context. When it was a particularly important novelty that sourdough can now also be found in dried form, it would maybe be good to explain this in an additional sentence.
5. Lines 33-36: Give specific references for all described effects and statements. Also remove the double spacing before “Beneficial”. Accordingly, check the whole manuscript for unnecessary double spacings (the search and replace function of word or other text-writing software can be used for this).
6. Lines 41-43: The aim is summed up well, but no clear focus on the probiotics is given – what should be expected from the title.
7. Line 48: Check if in this case, and also other parts of the manuscript, “back-slopped” is the right word or if it has to be replaced by “back-slopping”.
8. Lines 45-54: Concerning the classification of sourdough fermentation processes, a table is given. Nonetheless, it might be more appealing to present the reader a short overview figure instead.
9. Line 57-58: “The LAB species are naturally present in the flour, growing with a higher proliferation rate than yeast.” Can you provide any numbers from the literature about this (for example, specific growth rates)? Is it more like two or ten times higher? Are there any numbers stating how the populations differ throughout the process of fermentation?
10. Line 61: "LAB can consume pentose carbohydrates", which are the main pentoses present in sourdough fermentation processes (Xylose, Arabinose, others?)? It would be much appreciated if this information was added
11. Figure 1: Check the stoichiometry of the reactions, for example, if one or two lactate and ATPs molecules are generated during the homofermentative process. In the capture, remove “(2018)”. For the heterofermentative pathway, why is no fructose molecule displayed but written? For the homofermentative pathway, you give the molecule’s name and the structure, why only here?
12. Line 75: Should “Type II” here be replaced by “Type III”? What is the meaning of “stabilization” in this context? Does this mean the stability of the starter culture before it is added to the sourdough or does this concern the stability of the sourdough production process, which is improved by dried or lyophilized starters?
13. Table 1: Is the time period stated in the table for Type III correct (in line 77 it is said that it takes 24-72 hours)?
14. Line 89: “Sourdough microbiota”, why is the word capitalized here?
15. Lines 94-97: Do not break the line between numbers and units. Also, the sentence is difficult to understand. Consider rephrasing it or making two sentences.
16. Line 105: Is “in the environment's local” the correct expression here, or should it be “in the local environment”?
17. Line 113: What does “free pH” mean? If this refers to uncontrolled pH conditions during fermentations, please write it this way. Also, consider replacing “represents” with “yields”.
18. Lines 115-117: "A systematic analysis from 1990 to 2020 using “sourdough” as a keyword showed 115 that the sourdough microbiota is better studied in Europe, and Asia, specifically in China. In Latin America and America, few works are available." It seems that data were collected in this context and the statements derived from this. Please add the numbers for the geographical distributions of the studies.
19. Line 121: "..., almost from EUA," what is EUA? Is there a verb missing?
20. Line 134: Check if “prevail” has to occur twice.
21. Lines 142-143: "The predictive model showed a satisfactory result in predicting growth kinetic on microbial sourdough consortia." This topic and sentence lack a connection to the previous paragraphs. Please, also add what the “predictive” model is in this context.
22. Lines 152-155: “"Studies showed that using the spray drying technique to preserve the viability of lactic acid bacteria and yeasts in the dry mass was much more effective than observed in the freezing, drying, and freeze-drying techniques". Spray drying is a method of drying. Hence, the message of the sentence is unclear. Please rephrase.
23. Line 159: Check if “in scale” should be replaced by “at scale”.
24. Line 161: Replace “drier processes” by “drying processes”.
25. Table 4: Check the format of each column. Keep placing of thext in each column consistently (top, middle, bottom and left, centered, right). For example, reference 109 is not in one line with the others.
26. Lines 242-245: As a reader, I got the impression that the new chapter (4 Enzymes) starts with a general introduction to sourdough fermentation. This should not be necessary again when this was already covered in earlier manuscript parts. In fact, the new chapter could also start in line 246.
27. Line 252: Check if “each time” is correct in this context.
28. Figure 2: What does EPS mean? Add the explanation of the used abbreviation to the capture. Why is “Maltose” in blue letters and all other enzymes are not?
29. Lines 264-265: “It is an enzyme group that appears a metabolite of LAB" An enzyme cannot be considered a metabolite, which is known as an intermediate of a biochemical pathway. Please rephrase.
30. Line 281: “E.C.” Use the EC definitions throughout the entire manuscript consistent. Use “E.C.” or “EC” – but do not change between them. Also, ensure that spaces for classification are kept consistent (with or without space).
31. Line 298: “rheology properties” should be “rheological properties”.
32. Line 306: “in this fermentation metabolisms”. Please specify what “this” means here.
33. Line 308: Conser replacing “done” with “established”.
34. Line 395: “Triacylglycerol acyl hydrolase” – please decide if you capitalize enzyme names or not and keep it consistent within the manuscript.
35. Line 414: “The International Union of Biochemistry and Molecular Biology (IUBMB) classifies”, the “The International Union of Biochemistry and Molecular Biology” was already mentioned in the manuscript in line 317. Hence, introduce the abbreviation here and use IUBMB for later mentions in the manuscript.
36. Line 452: Add the American institution responsible for verifying GRAS status.
37. Line 464: Check if “that” is missing after “known”.
38. Lines 466-475: Since the situation in France is described, I was wondering if European Novel Food regulations have to be considered here. Or is sourdough fermentation excluded since it was conducted before 1997? However, would the addition of novel starter cultures change the production process so severely that the process would have to go through the Novel Food process? Please comment and complement.
39. Line 480: Consider replacing “singular” with “unique”.
40. Lines 480-482: “However, there are challenges to overcome to produce a suitable microbial starter or consortium on an industrial scale, maintaining its properties until bread-making methods.” This difficulty was not really addressed or explained in the manuscript.
41. Line 488: Add a space before “review”.
Author Response
Comments and Suggestions for Authors
In the reviewed manuscript, Akamine et al. elaborate on the use of probiotics in sourdough bread production. The article begins with a brief introduction, covering three short paraphs. Here, the importance of sourdough fermentation and the positive effects of probiotics are described. Finally, the authors state the aim of the article, which is "…to review the improvement of knowledge to use sourdough fermentation, its enzymes, metabolites, and technology advance to produce sourdough bread with desirable properties. "After the introduction, the article continues with a second chapter, giving an overview of common sourdough fermentation processes. Then, the article continues with a large third chapter, in which the sourdough microbiota is addressed. Here, one subchapter (3.1.) also addresses probiotics accordingly. Next, chapter four deals with enzymes' role in sourdough fermentation. The review is complemented by a fifth chapter, discussing regulatory issues concerning the use of microbe addition during sourdough bread production. Finally, the article closes with a short conclusion, stating that probiotics present in sourdough fermentation, as a fact, "promote improvements in quality, adding interesting nutritional, health, and sensorial characteristics, especially in bread. "But simultaneously, further studies have to be conducted to improve the understanding of microbes by omics technologies and synthetic biology, and also, it has to be ensured that the final food product is safe to use for the customers.
In general, the topic is very interesting and no other review article could be found that mainly addresses the subject of probiotics in sourdough fermentation. Nonetheless, the review should not be accepted to Fermentation (reject) in its current quality. However, I hope my comments below are helpful and will contribute to improving the quality of the manuscript.
General comments:
- The work's title suggests that the manuscript clearly focuses on probiotics in sourdough bread fermentation and gives an overview of the current research status on this topic. However, after reading the manuscript, I am not convinced that the exact effect of probiotics is the work's focus. Instead, it seems to be a more general review of sourdough fermentation. It is, of course, necessary to give information on related areas. However, it would have been much appreciated if these were more concise and
The review structure was modified to attend to the reviewer's request, and we created an improved section related to probiotics.
- Reading the article, I wondered why the authors had decided to structure it in its current form. Focusing on probiotics, it would have maybe been better to choose this as a single and major chapter and derive all other content and required background knowledge with current references from this. The present structure makes it hard to follow the article's key message. Also, although often containing interesting information, some parts of the manuscript do not appear to be well connected to previous parts – leading to some hard interruptions for the reader when working through the manuscript. For a review article, this should be improved.
All the review structure was modified to improve the quality of information, focusing on probiotics. The modifications were highlighted in the text.
- The quality of the manuscript is not sufficient – this does not only concern language but also some minor careless mistakes, which are not only a few. A manuscript submitted for peer review should be of higher quality. I hope my specific comments will also be helpful in this context.
Thank you for the corrections. A native English speaker editor revised the English, and all mistakes pointed out by the reviewer were corrected.
Specific comments:
- Line 15: The abstract mainly uses the present tense; therefore, change "were" to "are" to keep it consistent.
This correction has been made
- Line 22: Remove the comma before the reference "..,[1-3]".
This correction was made
- Line 28: Change "sensory food" to "food sensory".
This correction was made below, highlighted in yellow.
- Lines 28-30: "It is possible to find sourdough in the bakery industry, in liquid, in paste, or dry form with living microorganisms through the development of technologies, like spray drying, fluidized bed, or freeze-drying." The message of this sentence is not clear to me. The named drying technologies have been well-developed for decades, maybe "application" would be better in this context. When it was a particularly important novelty that sourdough can now also be found in dried form, it would maybe be good to explain this in an additional sentence.
Thanks for pointing out the error. Here we wanted to say that the industry can use sourdough in different formats. So we fixed it to: Sourdough can be used by the bakery industry in several applications through liquid, pasty or dry formats with live microorganisms.
- Lines 33-36: Give specific references for all described effects and statements. Also remove the double spacing before "Beneficial". Accordingly, check the whole manuscript for unnecessary double spacings (the search and replace function of word or other text-writing software can be used for this).
This correction was made
- Lines 41-43: The aim is summed up well, but no clear focus on the probiotics is given – what should be expected from the title.
As we answered in item 2, the review structure was modified to focus on probiotics.
- Line 48: Check if in this case, and also other parts of the manuscript, "back-slopped" is the right word or if it has to be replaced by "back-slopping".
This correction was made
- Lines 45-54: Concerning the classification of sourdough fermentation processes, a table is given. Nonetheless, it might be more appealing to present the reader a short overview figure instead.
According to the reviewer's suggestion, the table was replaced by a new figure.
- Line 57-58: "The LAB species are naturally present in the flour, growing with a higher proliferation rate than yeast." Can you provide any numbers from the literature about this (for example, specific growth rates)? Is it more like two or ten times higher? Are there any numbers stating how the populations differ throughout the process of Fermentation?
We added a new text to increase the understanding few articles in literature focused on this theme highlighted in the manuscript.
- Line 61: "LAB can consume pentose carbohydrates", which are the main pentoses present in sourdough fermentation processes (Xylose, Arabinose, others?)? It would be much appreciated if this information was added .
This new information has been added to the text.
- Figure 1: Check the stoichiometry of the reactions, for example, if one or two lactate and ATPs molecules are generated during the homofermentative process. In the capture, remove "(2018)". For the heterofermentative pathway, why is no fructose molecule displayed but written? For the homofermentative pathway, you give the molecule's name and the structure; why only here?
We replaced this figure and increased the information in the text about the fermentative process.
- Line 75: Should "Type II" here be replaced by "Type III"? What is the meaning of "stabilization" in this context? Does this mean the stability of the starter culture before it is added to the sourdough or does this concern the stability of the sourdough production process, which is improved by dried or lyophilized starters?
Thank you for the correction is type III. Type III can be dried or lyophilized. Stabilization concern to the production process. It is preferable for industrial bakery use because it has more quality stability than fresh sourdough.
- Table 1: Is the time period stated in the table for Type III correct (in line 77 it is said that it takes 24-72 hours)?
The time period of 24-72 hours is for Type II, according to (34). the correction was done in the text.
- Line 89: "Sourdough microbiota", why is the word capitalized here?
This correction was made.
- Lines 94-97: Do not break the line between numbers and units. Also, the sentence is difficult to understand. Consider rephrasing it or making two sentences.
This correction was made.
- Line 105: Is "in the environment's local" the correct expression here, or should it be "in the local environment"?
You are right, this correction has been made.
- Line 113: What does "free pH" mean? If this refers to uncontrolled pH conditions during fermentations, please write it this way. Also, consider replacing "represents" with "yields"..
This correction was made.
- Lines 115-117: "A systematic analysis from 1990 to 2020 using "sourdough" as a keyword showed 115 that the sourdough microbiota is better studied in Europe, and Asia, specifically in China. In Latin America and America, few works are available." It seems that data were collected in this context and the statements derived from this. Please add the numbers for the geographical distributions of the studies. I
The correction was done according to the reviewer's suggestion and the text was replaced as described below:
A systematic analysis from 1990 to 2020 using “sourdough” as a keyword showed the following number of articles by geographic location: North America (13), South America (6), Africa (31), Europe (175), Asia (54), China(20), and Oceania (1 - New Zealand).
- Line 121: "..., almost from EUA," what is EUA? Is there a verb missing?
This correction was made.
- Line 134: Check if "prevail" has to occur twice.
This correction was made.
- Lines 142-143: "The predictive model showed a satisfactory result in predicting growth kinetic on microbial sourdough consortia." This topic and sentence lack a connection to the previous paragraphs. Please, also add what the "predictive" model is in this context.
In the review process, the text was modified, and this part was replaced by:
‘The growth kinetics of six strains of LAB: Fructilactobacillus sanfranciscensis, were compared with five strains of yeast: Kazachstania humilis, from sourdough, by Altilia et al. 2021 [29]. Three predictive models were used to evaluate the behavior of co-cultivated microorganisms: The zwitweringg model based on Gompertz's equation, Baranyil and Roberts' function, and Schiraldi's function. The results showed that F. sanfranciscensis strains significantly steer the growth kinetics affecting the ratio of bacterial to yeast cells, the yeast strains of K. humilis adapt to the bacterial strains. The authors discuss the possibility of metabolic interactions for stabilizing the sourdough consortium through communication, like quorum sensing, to control population density, among other functionalities. In another co-culture article with LAB and yeast, the yeast population size decreased in the presence of LAB regardless of the strain, while the LAB´s population size was rarely influenced by the presence of yeast [30].In Type 0 fermentation, bioactive molecules and organic acids (lactic and acetic acids) are produced lowering the pH (pH~4), but it is controversial if this type fermentation is a true sourdough. The time limit is insufficient to produce other characteristic products of sourdoughs and is more known as sponge dough [27].”
- Lines 152-155: "Studies showed that using the spray drying technique to preserve the viability of lactic acid bacteria and yeasts in the dry mass was much more effective than observed in the freezing, drying, and freeze-drying techniques". Spray drying is a method of drying. Hence, the message of the sentence is unclear. Please rephrase.
This correction was made.
- Line 159: Check if "in scale" should be replaced by "at scale".
This correction was made.
- Line 161: Replace "drier processes" by "drying processes".
This correction was made.
- Table 4: Check the format of each column. Keep placing of thext in each column consistently (top, middle, bottom and left, centered, right). For example, reference 109 is not in one line with the others.
The tables were formatted correctly.
- Lines 242-245: As a reader, I got the impression that the new chapter (4 Enzymes) starts with a general introduction to sourdough fermentation. This should not be necessary again when this was already covered in earlier manuscript parts. In fact, the new chapter could also start in line 246.
the text was modified and replaced by the following:
“ Enzymes are responsible for several biochemical events in sourdough fermentation. Although some enzymes are present in the cereal, most are produced by microorganisms. From this perspective, the selection of native yeasts in sourdough for use in the baking industry has increased, and they are better adapted to improve their stability [165-166]. In the same way, bacteria, especially lactic acid bacteria, are widely selected for use in industrial bread-making [77]. “
- Line 252: Check if "each time" is correct in this context.
This correction was made.
- Figure 2: What does EPS mean? Add the explanation of the used abbreviation to the capture. Why is "Maltose" in blue letters and all other enzymes are not?
The corrections were made, and the figure was updated.
- Lines 264-265: "It is an enzyme group that appears a metabolite of LAB" An enzyme cannot be considered a metabolite, which is known as an intermediate of a biochemical pathway. Please rephrase.
This has been corrected, and the text has been rewritten.
- Line 281: "E.C." Use the EC definitions throughout the entire manuscript consistent. Use "E.C." or "EC" – but do not change between them. Also, ensure that spaces for classification are kept consistent (with or without space).
This correction was made.
- Line 298: "rheology properties" should be "rheological properties".
This correction was made.
- Line 306: "in this fermentation metabolisms". Please specify what "this" means here.
This correction was made.
- Line 308: Conser replacing "done" with "established".
This correction was made.
- Line 395: "Triacylglycerol acyl hydrolase" – please decide if you capitalize enzyme names or not and keep it consistent within the manuscript.
This correction was made.
- Line 414: "The International Union of Biochemistry and Molecular Biology (IUBMB) classifies", the "The International Union of Biochemistry and Molecular Biology" was already mentioned in the manuscript in line 317. Hence, introduce the abbreviation here and use IUBMB for later mentions in the manuscript.
This correction was made.
- Line 452: Add the American institution responsible for verifying GRAS status.
This correction was made.
- Line 464: Check if "that" is missing after "known".
The text has been rewritten.
- Lines 466-475: Since the situation in France is described, I was wondering if European Novel Food regulations have to be considered here. Or is sourdough fermentation excluded since it was conducted before 1997? However, would the addition of novel starter cultures change the production process so severely that the process would have to go through the Novel Food process? Please comment and complement.
the text was corrected according to the reviewer.
“In Europe, since 1997, with the Novel Food regulation, all food that started to be produce after 15 May 1997 is considered a novel food and has specific rules to be followed [271]. For microorganisms usually found in sourdough and used as a starter, it is challenging to define as a novel food because it sourdough predates 1997 [272]. Therefore, if the starter used in the sourdough fermentation is unusual or has some additional novel properties, it could be considered a novel food. For instance, UV-treated baker's yeast (Saccharomyces cerevisiae) has been considered a novel food since 2017 because it is alleged to increase vitamin D2 [273].Moreover, the EU1151/2012 recognizes that baker products could have a geographical denomination [11].”
- Line 480: Consider replacing "singular" with "unique".
This correction was made.
- Lines 480-482: "However, there are challenges to overcome to produce a suitable microbial starter or consortium on an industrial scale, maintaining its properties until bread-making methods." This difficulty was not really addressed or explained in the manuscript.
Thanks for the comment. All manuscript was modified to attend to the suggestions of the reviewers. The conclusion was also improved, and the questions of the reviewers were answered.
- Line 488: Add a space before "review".
This correction was made.
Round 2
Reviewer 1 Report
The authors have addressed the necessary comments. I am grateful to recommend it for publication as is it.Reviewer 2 Report
Significant amendments have been made, improving the quality of the article immensely. The authors' efforts are much appreciated, and the article might now be accepted for publication in its current form.